# Ballistic transport spectroscopy of spin-orbit-coupled bands in monolayer graphene on WSe$_2$

Qing Rao[1,6], Wun-Hao Kang[2,6], Hongxia Xue[1], Ziqing Ye[3], Xuemeng Feng[3], Kenji Watanabe[4], Takashi Taniguchi[5], Ning Wang[3], Ming-Hao Liu[2,7] ✉ & Dong-Keun Ki[1,7] ✉

Van der Waals interactions with transition metal dichalcogenides were shown to induce strong spin-orbit coupling (SOC) in graphene, offering great promises to combine large experimental flexibility of graphene with unique tuning capabilities of the SOC. Here, we probe SOC-driven band splitting and electron dynamics in graphene on WSe$_2$ by measuring ballistic transverse magnetic focusing. We found a clear splitting in the first focusing peak whose evolution in charge density and magnetic field is well reproduced by calculations using the SOC strength of ~ 13 meV, and no splitting in the second peak that indicates stronger Rashba SOC. Possible suppression of electron-electron scatterings was found in temperature dependence measurement. Further, we found that Shubnikov-de Haas oscillations exhibit a weaker band splitting, suggesting that it probes different electron dynamics, calling for a new theory. Our study demonstrates an interesting possibility to exploit ballistic electron motion pronounced in graphene for emerging spin-orbitronics.

The interfacial interactions with semiconducting transition metal dichalcogenides (TMDCs) have shown to be highly efficient in inducing strong spin-orbit coupling (SOC) in graphene[1–3]. For monolayer graphene, it was theoretically predicted to have two distinctive terms, one that couples out-of-plane spin and valley degrees of freedom (referred to as a spin-valley Zeeman term $\tau_z s_z$) and another that couples in-plane spin and sublattice degrees of freedom, similar to the Rashba term ($\tau_z \sigma_x s_y - \sigma_y s_x$), as follows:

$$H = H_0 + \Delta\sigma_z + \lambda\tau_z s_z + \lambda_R\left(\tau_z\sigma_x s_y - \sigma_y s_x\right), \quad (1)$$

where $H_0$ is the graphene's Dirac Hamiltonian, $\boldsymbol{\sigma} = (\sigma_x, \sigma_y, \sigma_z)$ is a Pauli matrix vector that acts on the sublattice degree of freedom in graphene, $\boldsymbol{s} = (s_x, s_y, s_z)$ is a Pauli matrix vector that acts on spin, and

$\tau_z = \pm 1$ identifies two different valleys in graphene ($\Delta \approx 0$ due to a large lattice mismatch between the graphene and TMDCs)[2,4–7]. Both SOC terms induce band splitting in graphene, and their strengths ($\lambda$ and $\lambda_R$) can be further tuned by an electric field perpendicular to the layers[8–13], twisting[14,15], or by pressure[16].

Combined with the high electron mobility and large experimental flexibility of graphene, such a strong interface-induced SOC makes the graphene on TMDCs ideal for ballistic spin-orbitronics where ballistic electron motion can be used to control or detect electron spin through the SOC[17–22]. It is particularly interesting as graphene has shown pronounced ballistic transport effects with large tunability, such as transverse magnetic focusing (TMF)[23–25], Veselago lensing[26,27], Fabry-Pérot interference[28,29], and ballistic snake states[30,31] among many others. They have also shown unique features originating from the

[1]Department of Physics and HK Institute of Quantum Science & Technology, The University of Hong Kong, Pokfulam Road, Hong Kong, China. [2]Department of Physics and Center for Quantum Frontiers of Research and Technology (QFort), National Cheng Kung University, Tainan 70101, Taiwan. [3]Department of Physics and Center for Quantum Materials, The Hong Kong University of Science and Technology, Clear Water Bay, Kowloon, 999077 Hong Kong, China. [4]Research Center for Electronic and Optical Materials, National Institute for Materials Science, 1-1 Namiki, Tsukuba 305-0044, Japan. [5]Research Center for Materials Nanoarchitectonics, National Institute for Materials Science, 1-1 Namiki, Tsukuba 305-0044, Japan. [6]These authors contributed equally: Qing Rao, Wun-Hao Kang. [7]These authors jointly supervised this work: Ming-Hao Liu, Dong-Keun Ki. ✉e-mail: minghao.liu@phys.ncku.edu.tw; dkki@hku.hk

relativistic nature of Dirac electrons[23–31]. However, a vast number of previous studies on graphene-TMDC heterostructures have focused on detecting spin relaxation due to electron scattering rather than the ballistic motion[1–3,7,16,32–37]. Moreover, only a few studies have found direct evidence for the SOC-induced band splitting by measuring beatings in Shubnikov-de Hass (SdH) oscillations (for both mono- and bilayer graphene)[3,10,38] or tracing changes in quantum capacitance (for bilayer graphene only)[11]. Not only to understand the effect of the SOC on the electronic properties of the system, such as the band topology[39,40] but also to exploit its full potential on ballistic spintronics[17–22], it is therefore essential to demonstrate the ballistic transport in graphene on TMDCs while simultaneously probing their band structures and electron dynamics.

To fill this missing link, we employ TMF technique in monolayer graphene on WSe$_2$ as it can not only probe the SOC-induced band splitting but also investigate electron dynamics simultaneously (see Fig. 1a–d). TMF occurs when ballistic carriers injected from a narrow aperture ("injector") at the edge of the sample are subject to a small perpendicular magnetic field ($\mathbf{B} = B\mathbf{z}$)[41–43]. Owing to the Lorentz force, the carriers follow skipping cyclotron orbits and focus on another narrow aperture ("collector") at a distance ($L$) that equals an integer

multiple of $2r_c$ with a cyclotron radius $r_c = \hbar k_F/eB$, where $\hbar$ is the reduced Planck constant, $k_F$ is the Fermi momentum, and $e$ is the elementary charge. Upon sweeping magnetic fields, the collector voltage will exhibit a set of resonance peaks at certain $B$-values determined by $k_F$,

$$B_j = \pm \frac{2j\hbar}{|e|L} k_F, \tag{2}$$

where $j$ is an integer and $\pm$ represents electron and hole for the configuration shown in Fig. 1b. This enables the detection of Fermi surface configurations[41–43]. In the systems with multiple bands[23], for instance, there will be multiple sets of resonance peaks at different $B$-values from which one can deduce their band structures. Moreover, TMF can also be used to study or control electron dynamics as charge carriers follow skipping cyclotron orbits during the process. In 2D electron gas (2DEG) systems with SOC, the TMF was indeed used to probe spin-orbit split bands and extract SOC strength by studying the separation of the peaks[17,19,21], deduce spin polarization by comparing their heights[17], or focus spin-polarized current by controlling ballistic electron motion[17,19,20].

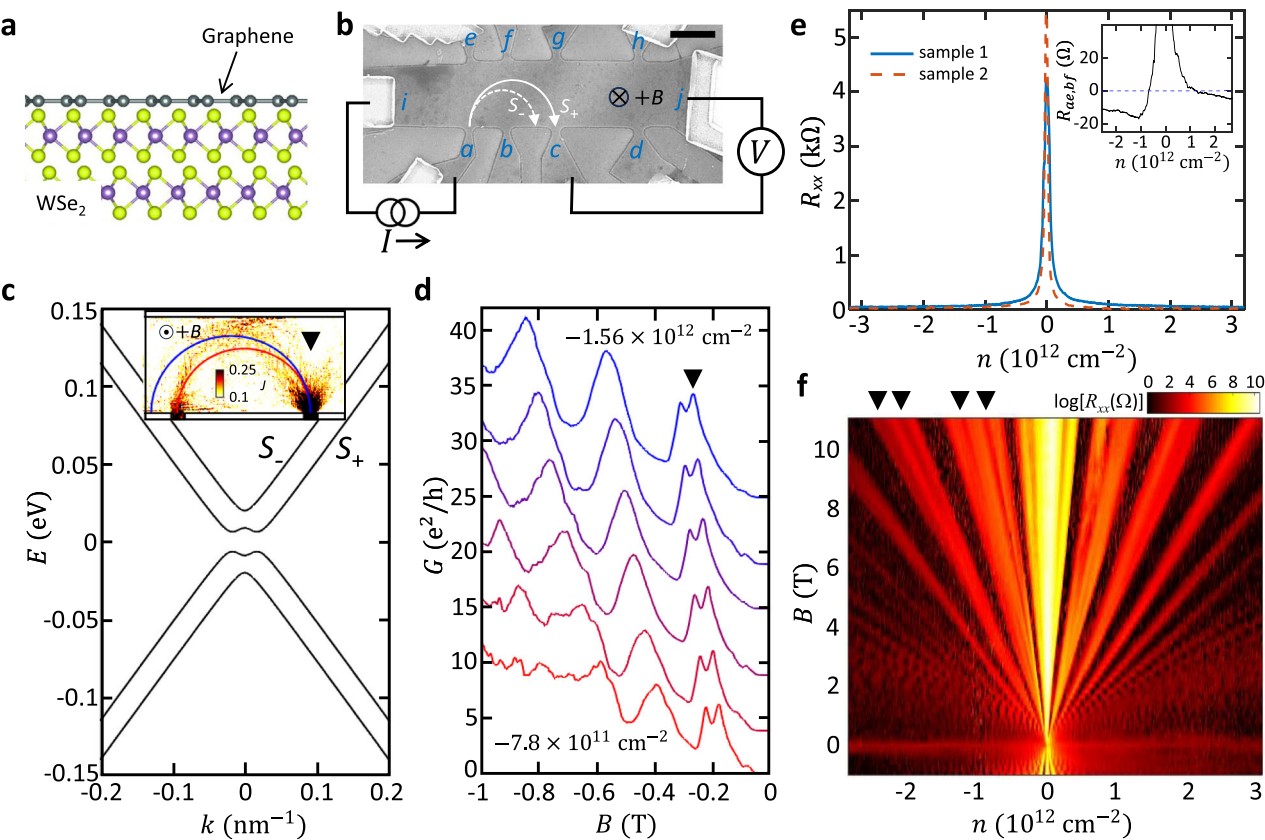

**Fig. 1 | Sample characteristics and TMF measurement scheme. a** The schematic of monolayer graphene-multilayer WSe$_2$ heterostructures. **b** Scanning electron microscope image of the device with a TMF measurement configuration (a scale bar: 2 μm; a distance between the injector $a$ and the collector $c$: $L \approx 4.0$ μm; a probe width: $w \approx 0.3$ μm). The two semicircles ($S_\pm$) illustrate trajectories of the carriers at different spin-orbit-coupled bands $S_\pm$ shown in **c** under perpendicular magnetic field $B$. **c** The energy dispersion of graphene on WSe$_2$ derived from the effective Hamiltonian Eq. (1) in the main text using SOC strengths of $\lambda = \lambda_R = 8.9$ meV ($\lambda_{SOC} \equiv \sqrt{\lambda^2 + \lambda_R^2} = 12.5$ meV). The inset shows the simulated local current density map at the focusing peak marked by a down triangle in **d**. **d** The corresponding TMF spectra, the conductance $G$ between the injector and collector as a function of magnetic field, calculated for an effective three-terminal device at 6 different hole densities, $n$ (in $10^{12}$ cm$^{-2}$) = −0.78, −0.93, −1.09, −1.24, −1.40, and −1.56 (from bottom

to top) using the tight-binding model. See Supplementary Note 1 for more details. **e** The carrier density ($n$) dependence of the four-probe resistances $R_{xx}$ of two different samples 1 (blue solid line) and 2 (red broken line) measured at 1.5 K. Both exhibit a sharp resistance peak at zero density, indicating high device quality. The inset shows the non-local Hall resistance $R_{ae,bf} \left( \equiv V_{bf}/I_{ae} \right)$, exhibiting a large negative signal on the hole side originating from the ballistic transport. **f** The Landau fan—the $\log(R_{xx})$ as a function of $n$ and $B$—plotted in a color scale (the darker color corresponds to the lower resistance), showing a high-quality quantum Hall effect measured at 1.5 K. On the hole side, the broken-symmetry states begin to appear at -3 T (indicated by black down-triangles), which indicates higher quality on the hole side, consistent with the large negative $R_{ae,bf}$ on the hole side shown in the inset of **e**.

Here, we demonstrate that all these studies are possible in graphene-TMDC heterostructures. We first show a clear splitting of the first focusing peak in graphene on WSe$_2$, whose evolution in charge density and magnetic field matches well with the theoretical calculations shown in Fig. 1c, d using the SOC strength of -13 meV (see Supplementary Notes 1,2 for simulation details[44,45]) and no splitting in the second peak, which indicates a stronger Rashba SOC in the system. From the temperature dependence measurement, we also find a possible suppression of electron-electron scatterings that may originate from the dielectric scattering of WSe$_2$ layers and/or the induced SOC. Further, we show that Shubnikov-de Haas oscillations exhibit a weaker band splitting, suggesting that it probes different electron dynamics. Interestingly, a similar behavior was found in studies on 2DEG with SOC[19,21], indicating that this is universal and a new theory is needed to explain the phenomenon. Our study, therefore, places graphene-TMDC heterostructures as an interesting new material platform to explore the effect of SOC on ballistic electron transport in low-dimensional systems.

## Results

### Sample characterization

To study the ballistic TMF effect, we used a dry pick-up and transfer technique to assemble a stack of hexagonal boron-nitride (hBN), monolayer graphene, and multilayer WSe$_2$ such that the graphene is protected from the harsh chemical environments in the following nanofabrication process[46]. Standard electron beam lithography and lift-off were carried out to make Hall bar devices on 285-nm-thick SiO$_2$ substrate with doped silicon underneath used as a gate to control charge density $n$ (see Fig. 1a, b and Methods for fabrication details). Electron transport through the fabricated devices was measured in a 1.5 K variable temperature insert with a 14-T superconducting magnet using a standard low-frequency AC lock-in technique (see Methods). Classical and quantum Hall effect measurements were used to estimate gate capacitance and confirm that the graphene flakes in our samples are monolayers (see Supplementary Fig. 1). Figure 1e shows the carrier density dependence of the four-probe resistances of two different samples 1 and 2 measured at 1.5 K, exhibiting high quality with carrier mobility of 200,000 ~ 400,000 cm$^2$V$^{-1}$s$^{-1}$. Especially on the hole side, the symmetry broken quantum-Hall states were observed at magnetic fields as low as ~3 T (marked by black down-triangles in Fig. 1f). This indicates higher hole mobility in our sample than on the electron side. It is also consistent with the observation of the larger negative non-local Hall resistance $R_{ae,bf}$ on the hole side (the inset of Fig. 1e) originating from the ballistic electron motion[47] ($R_{\alpha\beta,\gamma\delta} \equiv V_{\gamma\delta}/I_{\alpha\beta}$ which refers to the resistance measured by sending a current from contact $\alpha$ to $\beta$ and measuring the voltage between contacts $\gamma$ and $\delta$). Having such a high mobility—equivalently, a long mean free path—is important to resolve the small splitting of the focusing peak expected theoretically (Fig. 1d).

### Transverse magnetic focusing spectra

The TMF signal ($R_{nl} = V_{cj}/I_{ai}$) is measured in a non-local configuration upon varying $n$ and $B$, as depicted in Fig. 1b. Figure 2a, b show the resulting maps of $R_{nl}(n,B)$ from sample 1 and 2, respectively. Both exhibit similar TMF spectra and their evolutions in $n$ and $B$. Overall, the positions of the $j$-th TMF peak in $B$ follow Eq. (2) with $k_F = \sqrt{\pi|n|}$, as expected for monolayer graphene. However, on the hole side (where we found the higher sample quality), we can identify the splitting of the first focusing peak that evolves continuously in $n$ and $B$ and no splitting in the second (here, we only focus on the peaks that appear in all density range). Figure 2c, d further magnify the features by plotting 1D cuts $R_{nl}(B)$ of the map at different $n$, which qualitatively matches the simulation result shown in Fig. 1d (see Supplementary Fig. 2a for the 1D cuts of the map on the electron side for comparison). Moreover, different from the TMF measurements on pristine monolayer graphene[23],

we found a large TMF signal constantly exceeding 100 Ω around zero density (nearly two orders of magnitude larger than the values at finite densities; see the dark red bands near zero density in the color maps shown in Fig. 2a, b). All features found in the experiment (Fig. 2) match well with our expectations for the graphene with SOC and provide valuable insights about microscopic electron processes in the system, as discussed below.

### Analysis of the first focusing peak

First, the first focusing peak splits due to the SOC-induced multiple bands $S_+$ and $S_-$ in our sample (Fig. 1c). Note that such a prolonged splitting in both $n$ and $B$ has not been observed in other graphene systems without SOC[23–25]. Moreover, we were able to fit the positions of the first focusing peaks with calculations using the SOC strength of $\lambda_{SOC} \equiv \sqrt{\lambda^2 + \lambda_R^2} = 13.9$ meV and 12.1 meV for samples 1 and 2 respectively as marked by black dotted lines in Fig. 2a, b. Figure 3a and Supplementary Fig. 2b further emphasize the accuracy of the fitting by plotting the average of the normalized difference between the data and the calculation $\langle \delta B^2 \rangle$ as a function of $\lambda$ and $\lambda_R$ in a color scale for devices 1 and 2, respectively (a darker color indicates the smaller $\langle \delta B^2 \rangle$ so the better fitting; see the caption for more details). We note that the fitting works for any values of $\lambda$ and $\lambda_R$ as long as they satisfy $\lambda_{SOC} = 13.0 \pm 4.7$ meV for both samples, indicating that the $\lambda$ and $\lambda_R$ have a similar effect on the splitting of the first TMF peak—equivalently, the band splitting—in the density range we studied. This is expected as in the density range explored, the Fermi energy is larger than both $\lambda$ and $\lambda_R$, leading to an identical splitting in momentum, $\Delta k = 2\sqrt{\lambda^2 + \lambda_R^2}/\hbar v_F$ with Fermi velocity $v_F \approx 10^6$ m·s$^{-1}$ for both $\lambda$ and $\lambda_R$. This is consistent with the previous SdH oscillations measurements that probe electron bands[3,38].

In addition, we found that the amplitude of the first split peak closer to zero $B$ is always lower than that of the second (see, e.g., Fig. 2c, d). Interestingly, in 2DEG with Rashba SOC, such uneven heights of the split peaks have been used as a signature of the spin polarization in the system, even though more rigorous analysis and more controlled experiments are needed to identify the origin of the spin polarization[17,48–51]. For instance, it was suggested that adiabatic transition between the quantized sub-bands formed at the injector with width $w$ could polarize electron spin[48,51]. Slightly modifying the condition derived for 2DEG[51], we get $\frac{\Delta k}{2k_F} > \frac{3}{16}\left(\frac{\lambda_F}{w}\right)^2 \rightarrow \lambda_R > \frac{3\pi}{8}\frac{\hbar v_F \lambda_F}{w^2} \approx 0.25 \sim 0.35$ meV for our sample with Fermi wavelength $\lambda_F \approx 30 \sim 40$ nm, and $w \approx 300$ nm. The SOC strengths estimated in our study are well in the range (Fig. 3a), but due to the absence of the quantum point contacts in our device, we cannot confirm the formation of the sub-bands at the injector. On the other hand, it was also shown that the relative heights of the split peaks could vary with the distance between the injector and detector due to the difference in scattering lengths for carriers at different spin-orbit-split bands[50] and that the exact momentum dependence of the SOC should be considered to understand the spin-polarization[49]. Interestingly, in graphene on TMDCs, the presence of both spin-valley Zeeman and Rashba SOC terms was predicted to induce a characteristic spin winding of the spin-orbit-coupled bands that leads to a current-induced spin polarization[52], which may result in the uneven heights of the peaks. For better understanding, we will need to conduct more sophisticated experiments, such as applying in-plane magnetic fields to control Zeeman energy[17,49,53], using samples with various distances between the injector and detector[50], or using ferromagnetic contacts for spin-sensitive detection[54]. Nevertheless, we note that all these previous studies have used the uneven heights of the split peaks as a signature of the spin polarization in the system[17,48–51]. Thus, our study shows a

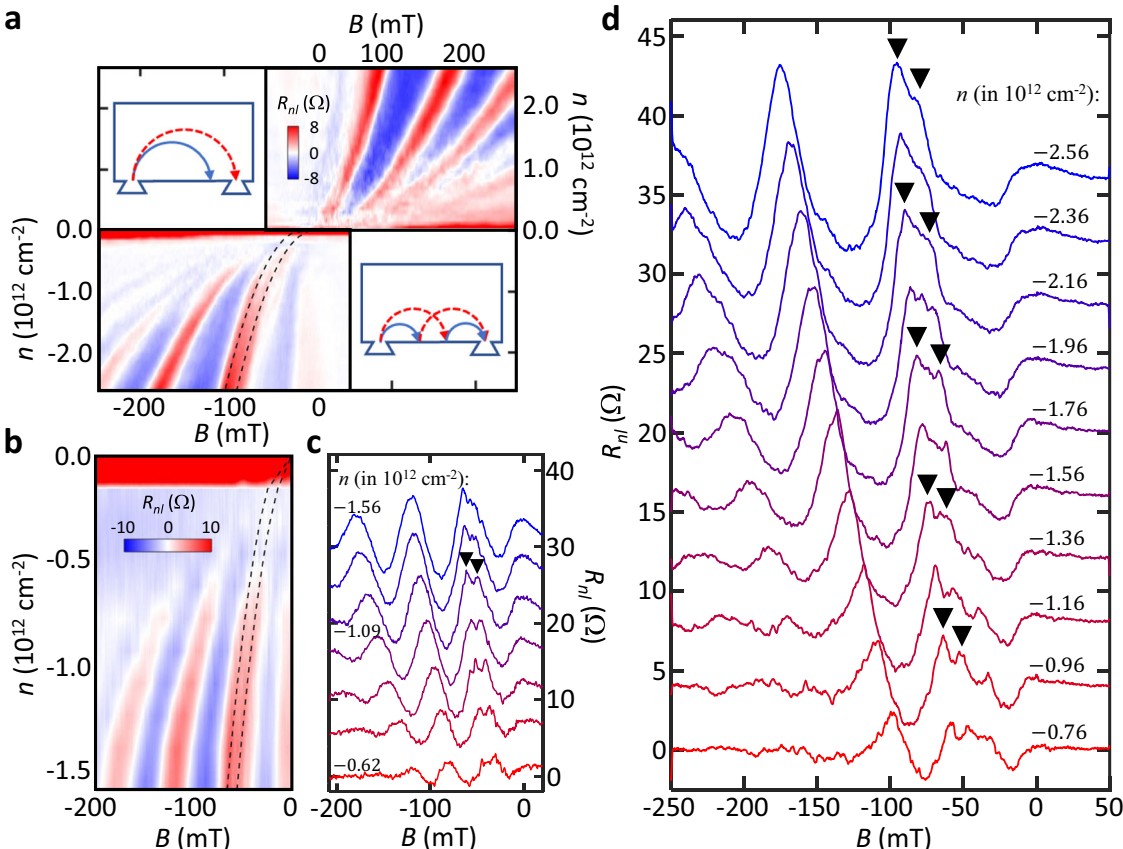

**Fig. 2 | TMF spectra. a** Color-scale maps of TMF signal $R_{nl}(B,n)$ measured in sample 1 at 1.5 K (top right: electron side; bottom left: hole side). The broken lines show the theoretically calculated focusing peaks at $\lambda_{SOC} = 13.9$ meV. Inset: carrier trajectories for the first and second focusing peaks (top left and bottom right, respectively). **b** TMF map and **c** the corresponding 1D cuts measured in sample 2 at 1.5 K at $n$ (in $10^{12}$ cm$^{-2}$) = −0.62, −0.78, −0.93, −1.09, −1.24, −1.40, and −1.56 (from bottom to top). The broken line shows the theoretically calculated focusing peaks at $\lambda_{SOC} = 12.0$ meV. **d** 1D cuts of the data from sample 1 shown in **a**. The black down-triangles in (**c**) and (**d**) mark some of the two split peaks for guidance.

possibility of using TMF to detect spin polarization of the ballistic carriers in graphene-TMDC heterostructures.

## Analysis of the second focusing peak

The second key finding of this study is the absence of the splitting in the second focusing peak (Fig. 2), which provides more information about the nature of the SOC in the system. It first can be interpreted as the scattering of charge carriers between the spin-orbit-coupled bands $S_\pm$ at the sample edge, which leads to a single peak as illustrated in the bottom inset of Fig. 2a. To confirm this origin, we have further calculated electron trajectories for the second peak with or without the inter-band transition in the sample in Supplementary Fig. 3. As expected, without the inter-band mixing, the second focusing peak also exhibits splitting. This confirms that the absence of the splitting in the second peak originates from the scattering between the bands $S_\pm$ at the sample edge.

Interestingly, from the behavior of the second peak, we can learn more about the relative strength of the spin-valley Zeeman and Rashba SOC terms, $\lambda$ and $\lambda_R$, because the inter-band scattering at the edge depends sensitively on the spin textures of the split bands $S_\pm$. For more accessible discussion, let us first consider a single valley only and discuss the effect of the intervalley scattering later. As depicted in Fig. 3b, c, when only spin-valley Zeeman term exists (in other words, when $\theta_{SOC} = 0$; see Fig. 3a), spins in $S_+$ ($S_-$) band are aligned up (down) in z-direction. Thus, when backscattered at the edge, the electron at state A in the band $S_+$ will jump to state B in the same band unless there are enough magnetic impurities to flip the spin, which is unlikely in high-quality graphene samples like ours. This would lead to the

splitting of the second focusing peak as illustrated in Fig. 3b. On the other hand, when the Rashba term dominates (i.e., when $\theta_{SOC} = \pi/2$), $S_\pm$ bands have an opposite spin winding such that the electron at the state A in the band $S_+$ will jump to the state C in the opposite band, leading to the merging of the peaks as depicted in Fig. 3d, e. Therefore, we can estimate that in our system, the Rashba term dominates. Similarly, studies on 2DEG systems with Rashba SOC have indeed shown no splitting in the second focusing peak[19,21,55].

To further confirm our analysis, we have simulated TMF spectra for different values of $\theta_{SOC}$ in Fig. 3f. As shown in the figure, the splitting in the second peak disappears rapidly as $\theta_{SOC}$ increases from zero and becomes nearly invisible as $\theta_{SOC} \gtrsim \pi/4$, consistent with our analysis above. It is, however, worth mentioning that in the simulation, we used an ideal edge, so we may have underestimated the intervalley scattering probabilities that occur in the real sample edge with atomic defects[56]. Although this does not influence the Rashba-dominating case as the spin winding direction remains the same for the $S_\pm$ bands in different valleys (see Fig. 3e), it can affect the result when the spin-valley Zeeman term dominates because the spin orientation for each $S_\pm$ band becomes opposite in different valleys (see Fig. 3c). Thus, the intervalley scattering can lead to the scattering from state A at K valley to state C' at K' valley, i.e. the backscattering between the $S_\pm$ bands at the edge, when the spin-valley Zeeman term dominates, suppressing the splitting in the second peak. Although more experimental and theoretical works are required for a complete understanding of this feature, we can roughly assume that our sample has disordered edges with atomic scale defects with resonance energy near the charge neutrality[57], leading to the intervalley scattering rate close to or less

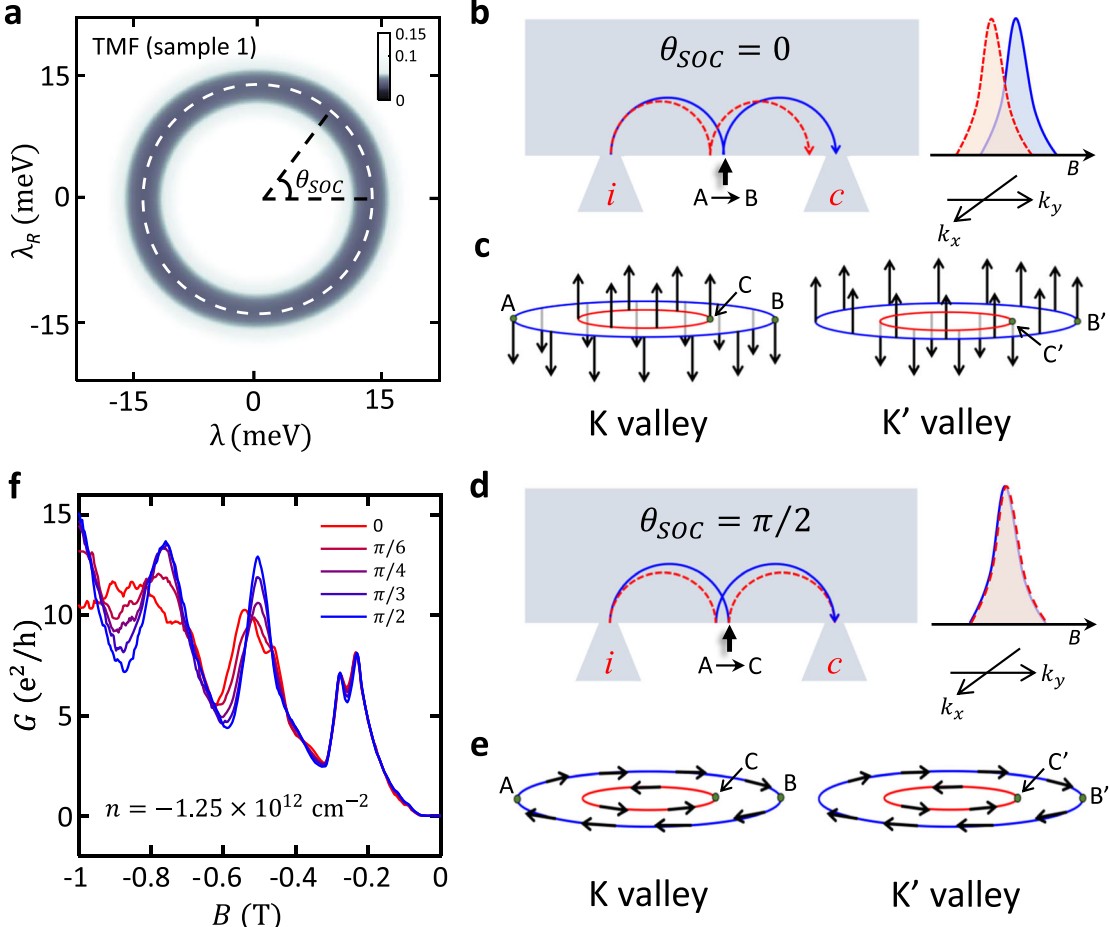

**Fig. 3 | Analysis of TMF signal. a** The color-scale map of the average difference $\langle \delta B^2 \rangle \equiv (\sum[(\Delta B_+/B_0)^2 + (\Delta B_-/B_0)^2])/N$ as a function of $\lambda$ and $\lambda_R$ from sample 1 ($N$: number of data used). $\Delta B_\pm$ is the difference between the predicted focusing peak positions from the simulation for certain ($\lambda$, $\lambda_R$) and the real peak positions measured in the experiment for the band $S_\pm$, whereas $B_0$ is the half of the maximum splitting observed. Thus, the smaller $\langle \delta B^2 \rangle$ (darker in the map) indicates a better agreement. A dashed white circle draws the best-fit value. We use a criterion $\langle \delta B^2 \rangle \leq 0.1$ to extract the SOC strengths of $\lambda_{SOC} = 13.9 \pm 4.0$ meV (and $12.0 \pm 3.5$ meV for sample 2, see Supplementary Fig. 2b). $\theta_{SOC}$ in **a** is defined as $\cos^{-1}(\lambda/\lambda_{SOC})$. **b**–**e** Comparison of the electron trajectories for the second focusing peak (in the absence of the intervalley scattering) and spin configurations for the two cases when there is only the spin-valley Zeeman term (**b, c**; $\theta_{SOC} = 0$) and when

Rashba term exists (**d, e**; $\theta_{SOC} = \pi/2$). The shapes of the resulting focusing peaks for each case are shown in the inset of **b** and **d**. Without intervalley scattering, due to the spin conservation, the electron at the edge (state A) will be backscattered to B when $\theta_{SOC} = 0$ (**b, c**), leading to the splitting in the second peak, whereas when $\theta_{SOC} = \pi/2$, it will be transferred to state C (**d, e**). When the intervalley scattering is present, the backscattering from state A to C' can occur even for the $\theta_{SOC} = 0$ case (**c**), leading to the suppression of the splitting in the second peak. See the main text for details. **f** The calculated TMF spectra with varying $\theta_{SOC}$ at $n = -1.25 \times 10^{12}$ cm$^{-2}$ when the overall SOC strength $\lambda_{SOC} = 10$ meV. One can clearly see that the positions of the first focusing peaks remain the same while the second peak shows multiple peaks near $\theta_{SOC} = 0$.

than that of the intravalley scattering in the density range explored. In this case, we would still expect to see the splitting when $\theta_{SOC} = 0$. Thus, we believe that the absence of the second peak splitting in Fig. 2 indicates the stronger Rashba SOC in our system.

**Large non-local resistance and temperature dependence**

We can also explain the observed large $R_{nl}$ near zero density (Fig. 2a, b) as the presence of the spin Hall effect (SHE)[1] in the system. From the very weak temperature dependence of the conductance minimum at zero density (Fig. 4a), we first confirm that this is not from the gap opening at charge neutrality, which may have given a large TMF signal as found in the gapped trilayer graphene[23]. In contrast, we found a large non-local Hall signal $R_{nl}^H = R_{ae,cg}$ near the charge neutrality that exceeds the ohmic contribution by about 230 Ω (Fig. 4b; here we used Hall probes that are further apart from those used in the inset of Fig. 1e to reduce the influence from the ballistic negative resistance). This is consistent with the SHE found in a similar system[1] previously, except that the signal appears near zero density below $2.0 \sim 3.0 \times 10^{11}$ cm$^{-2}$ in our device. We attribute this to the crossover from the diffusive

regime, where the SHE occurs, to the ballistic regime, where the TMF effect appears. In fact, we found that the density $2.0 \sim 3.0 \times 10^{11}$ cm$^{-2}$ coincides well with the value above which the ballistic negative non-local resistance (the inset of Fig. 1e) and TMF signals appear (Fig. 2 and Supplementary Fig. 4) within the resolution of our experiment. More in-depth studies on the crossover between the diffusive spin-Hall and ballistic TMF effects or their coexistence may lead to a better understanding of the charge transport in spin-orbit-coupled systems, and our study shows that it is possible in high-quality graphene-TMDC heterostructures.

Additionally, the observation of both diffusive SHE at low density and ballistic TMF peak splitting at higher density indicates that the SOC in our sample is induced by proximity with TMDC[2,4–7] not by defects[1] as the defects would have strongly suppressed ballistic transport in the sample. It confirms that in graphene-TMDC heterostructures—thanks to the atomically sharp interface—the atomic potentials generated by the TMDC can influence graphene band strongly to create an effective Hamiltonian with distinctive SOC terms shown in Eq. (1)[2,4–7]. This is similar to how the atomic potentials of

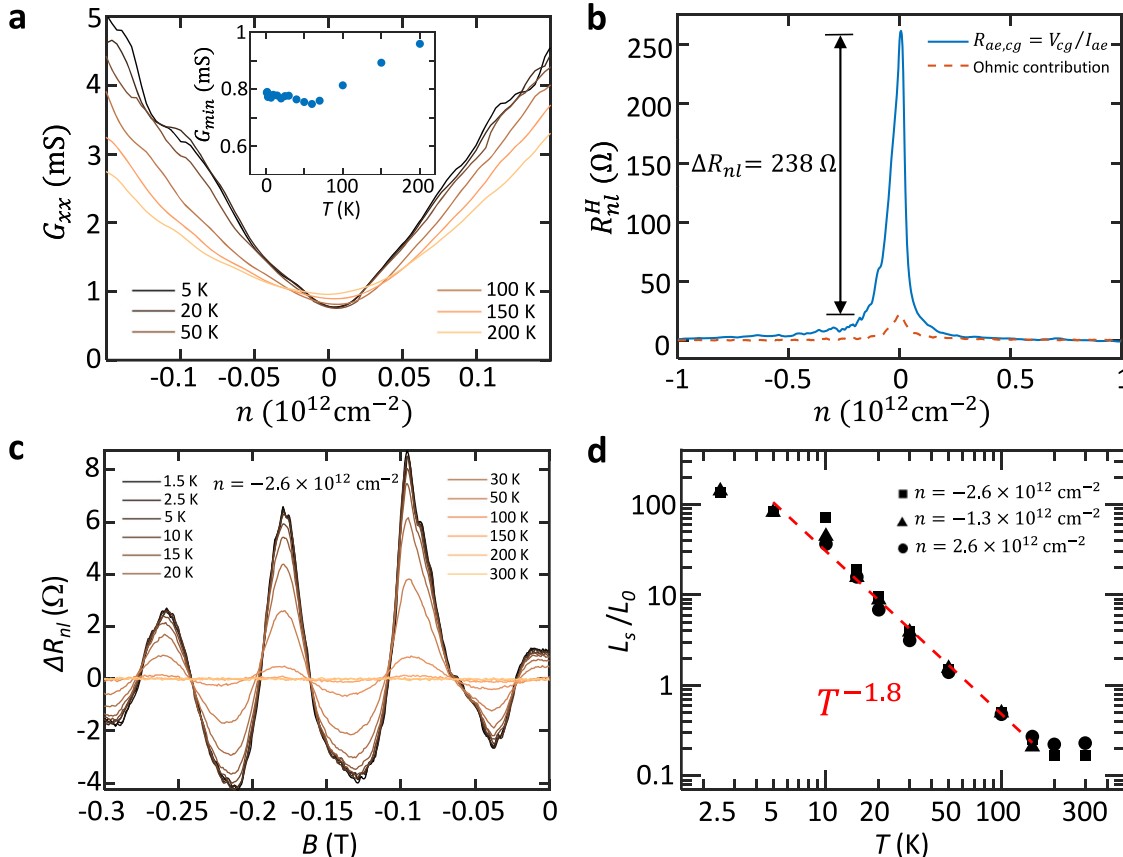

**Fig. 4 | Temperature dependence measurements. a** Temperature dependence of the local four-terminal conductance $G_{xx} = 1/R_{xx}$ as a function of charge density, exhibiting a weak temperature dependence of the minimum conductance $G_{\min}$ at zero density as magnified in the inset. **b** Non-local Hall resistance $R_{ae,cg} = V_{cg}/I_{ae}$ as a function of carrier density $n$ (solid blue line) compared with the calculated Ohmic contribution (broken red line), consistent with the spin Hall effect. **c** Temperature dependence of the TMF spectra at $n = -2.6 \times 10^{12}$ cm$^{-2}$. The smooth backgrounds are extracted by a Gaussian filter with a full width at half maximum of 0.2 T, which is larger than the oscillation period of TMF signals. **d** The relative scattering lengths (calculated from areas below the first focusing peaks) at $n = -2.6 \times 10^{12}$ cm$^{-2}$ (square), $n = -1.3 \times 10^{12}$ cm$^{-2}$ (triangle), and $n = 2.6 \times 10^{12}$ cm$^{-2}$ (circle) as a function of temperature plotted in a log scale, which follows the $T^{-1.8}$ dependence, indicated by the dashed red line. See the main text for more discussions.

boron and nitrogen atoms in hBN creates the moiré minibands in graphene-hBN moiré structure[58–61]. We note that such a proximity effect does not require charge carriers in graphene to fill the energy bands in TMDC (or hBN) and thus it can occur even when there are no defect sites in TMDC that can sink charge carriers from graphene and suppress the ballistic transport. Our observation also aligns well with other studies on similar graphene-TMDC heterostructures[2,3,7,16,32–38,62].

We now examine the temperature dependence of the TMF spectra in Fig. 4c, d to study the electron dynamics in the system. Upon increasing temperature, we found that the amplitude of the TMF spectra decreases (Fig. 4c). This suggests enhanced electron scattering at high temperatures. To identify the main scattering mechanism in our system, we extracted the total area below the first focusing peaks $A_1$ at varying temperatures from 1.5 K to 300 K and calculated the relative scattering length from $L_S/L_0 = (\ln[(A_1(1.5K)/A_1(T)])^{-1}$, which is proportional to the effective scattering time ($L_0$ is the length of the semi-circular electron trajectory corresponding to the first focusing peak)[24,25]. Figure 4d shows the result exhibiting a clear $T^{-1.8}$ dependence on both electron and hole side for different charge densities. This is between electron-phonon scattering ($T^{-1}$) and electron-electron ($e$-$e$) scattering ($T^{-2}$), indicating that although the $e$-$e$ scattering is dominant, it is also slightly suppressed in our sample. In comparison, similar TMF studies on graphene-hBN heterostructures[24,25] have shown $T^{-2}$ dependence in a wide range of temperatures and charge density. Thus, the $T^{-1.8}$

dependence found in our sample should be from the WSe$_2$ itself and/or the induced SOC. Although the exact origin is unclear, we note that the WSe$_2$ has a dielectric constant ($\varepsilon_0 \approx 7.9$) about twice larger than hBN used in the previous studies ($\varepsilon_0 \approx 3.8$)[63] that may induce more screening. In addition, recent studies[64,65] have shown that the SOC can affect electronic interaction phenomena such as superconductivity in twisted or Bernal bilayer graphene. This suggests a possibility to measure ballistic transport effects in graphene-TMDC heterostructures to study the effect of SOC on $e$-$e$ or electron-hole interaction phenomena, such as viscous charge transport[66–68], and electron-hole collisions[69], or superconductivity[64,65].

### Comparison with Shubnikov−de Haas (SdH) oscillations

To further elucidate the band splitting in our system, we have measured SdH oscillations at a higher magnetic field range. The results are summarized in Fig. 5. In all the density ranges including the electron side, we found beatings in the oscillations originating from the spin-orbit-coupled split bands (Fig. 5b). For quantitative analysis, we performed fast Fourier transforms (FFT) and extracted the frequency $f$ at which the spectra exhibit a peak (Fig. 5a) which is directly connected to the area of the corresponding Fermi surface by $f = nh/2e$ and $n = k_F^2/2\pi$ assuming broken spin degeneracy due to SOC. It can therefore be used to estimate the SOC strengths independently. Figure 5c shows the result (we have selected the peaks that evolve

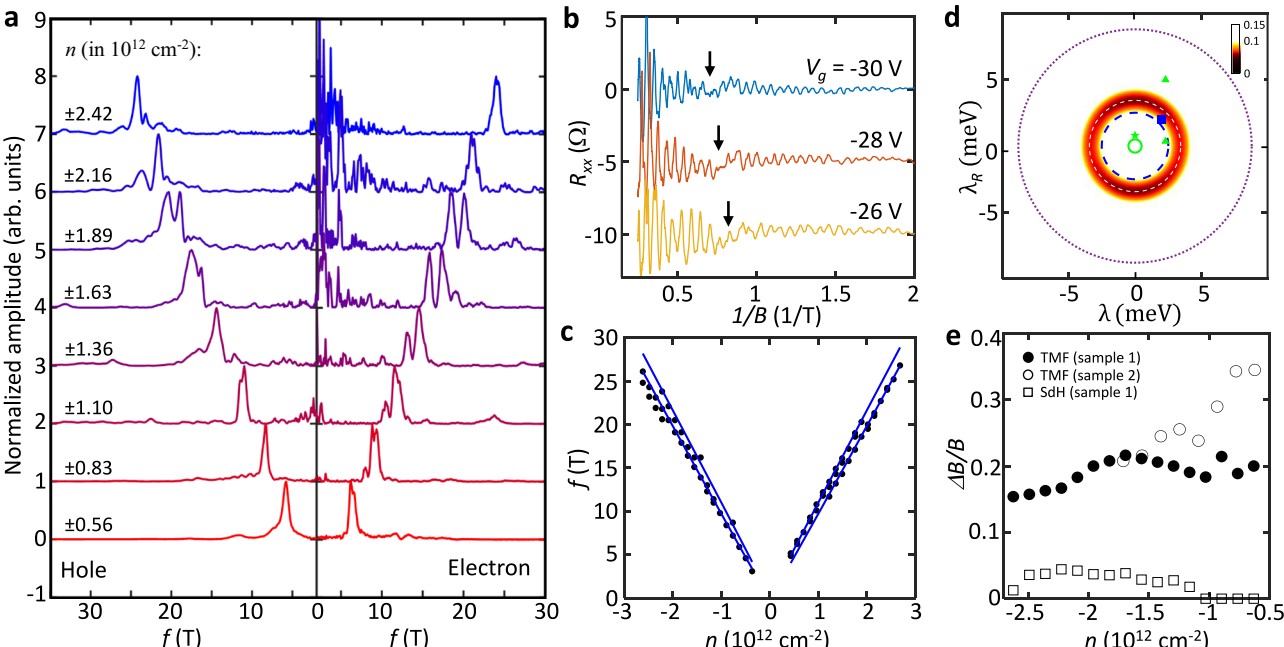

**Fig. 5 | SdH oscillations. a** The FFT spectra derived from the SdH oscillation curves as a function of carrier density $n$, for both electron and hole side. **b** Representative SdH oscillation curves, which clearly show the beating patterns (marked by arrows; curves are vertically offset for clarity). **c** The frequency peak positions extracted from the FFT spectra at different carrier densities. The solid lines show the fitting with calculated band structures using $\lambda_{SOC} = 3.4$ meV. **d** The color-scale map of the average difference $\langle \delta B^2 \rangle$ in $\lambda$ and $\lambda_R$ from the SdH oscillations with the best fit value drawn by the dashed white circle (see the caption of Fig. 3a for details). Using the same criterion used in Fig. 3a $\langle \delta B^2 \rangle \leq 0.1$, we get $\lambda_{SOC} = 3.4 \pm 0.7$ meV. Note that around the circle near $\lambda = 0$, the color becomes darker, indicating stronger Rashba SOC in the system consistent with discussions in Fig. 3b-f. The values from the

previous studies are included for comparison: a dotted purple circle from spin Hall effect[1], a dashed blue circle from SdH oscillations[38], a square from Landau level splitting[62], and circle[32], star[33], and two filled green triangles[37] (two different values are obtained from different spin relaxation mechanisms) from weak anti-localization measurements. Some studies[1,32,33,62] used $\lambda_R/2$ and/or $\lambda/2$ in the Hamiltonian Eq. (1), so we divided the values by half in the plot. **e** The normalized splitting $\Delta B/B$ of the first TMF peak ($\Delta B_{TMF}/B_{TMF}$) at different charge densities extracted from Fig. 2 compared with the splitting calculated from the SdH oscillations ($\Delta B_{SdH}/B_{SdH}$). $\Delta B_{TMF}$ and $\Delta B_{SdH}$ are the sizes of splitting in $B$, while $B_{TMF}$ and $B_{SdH}$ are the averaged peak positions of two sub-peaks. Over the whole density range, the $\Delta B_{SdH}/B_{SdH}$ remains smaller than the $\Delta B_{TMF}/B_{TMF}$.

continuously in density only), which offers a good agreement with the calculation using the SOC strengths along the circle $\sqrt{\lambda^2 + \lambda_R^2} = 3.4 \pm 0.7$ meV (Fig. 5d). Interestingly, the fitting is slightly better (i.e., $\langle \delta B^2 \rangle$ is smaller) near $\lambda = 0$ which indicates the larger Rashba SOC in the system, consistent with our estimation from the analysis on the second focusing peak (Fig. 3b–f). The absolute value is, however, about 4 times smaller than those extracted from the TMF data (Fig. 3a and Supplementary Fig. 2b), indicating that the TMF and SdH oscillations probe different electron dynamics.

To compare the results more directly, we have calculated $\Delta B_{SdH} = (2\hbar/|e|L)|k_{F1} - k_{F2}|$, the expected size of the splitting of the first TMF peak in $B$ using the $k_F$-values extracted from the SdH oscillations (Fig. 5c), and plotted the splitting measured in TMF, $\Delta B_{TMF}$ (extracted from Fig. 2), together in Fig. 5e after normalizing the values with the averaged peak positions of the two sub-peaks. As shown in the figure, $\Delta B_{TMF}/B_{TMF}$ remains larger than $\Delta B_{SdH}/B_{SdH}$ in all density range accessed in the experiment. Interestingly, we found that similar behavior was observed in 2DEG systems with SOC[19,21], where it was suggested[21] that there might be a SOC term, such as a linear-$k$ term, that does not affect the total area of the Fermi surface by shifting a circle in one momentum direction. Since the SdH oscillation requires carriers to complete a full cyclotron motion while in TMF, they only make a half turn, it may be possible that one finds larger splitting in TMF than in SdH, as seen experimentally. However, it is also possible that the SdH oscillations require relatively large magnetic fields to form Landau levels which can induce non-negligible Zeeman energy and may affect the spin-valley Zeeman and Rashba terms in Eq. (1) differently (see Supplementary Note 3 for more discussions). Although more studies are required to understand this discrepancy, our

measurement shows that the behavior occurs not only in semiconductor heterostructures-based 2DEG systems[19,21] but also in graphene when SOC exists. Therefore, there is likely a fundamental origin behind this phenomenon.

## Comparison with previous studies

In Fig. 5d, we also include all SOC strengths extracted from the previous measurements on monolayer graphene-TMDC heterostructures for comparison. Overall, the relaxation time analysis from weak anti-localization or spin-Hall effect measurements[1,32,33,37] shows a considerable sample-to-sample variation (Fig. 5d). It can be from the fact that these measurements rely on the model to connect the spin relaxation process in the system with the SOC strength, which is sensitive to the sample-specific electron scattering process[7,37]. On the other hand, TMF and SdH oscillations directly probe the size of the Fermi surface, which can be compared with theoretically calculated band structures without considering details of the scattering processes. The recent study on SdH oscillation in monolayer graphene-$WSe_2$ heterostructures[38] has indeed shown a SOC strength $\lambda_{SOC} = 2.51$ meV close to ours (a dotted circle in Fig. 5d). Interestingly, the study on Landau level splitting[62], which is closely related to the SdH oscillations, also showed a similar SOC strength (a square in Fig. 5d). Moreover, in our TMF study, we found similar SOC strengths in two different samples (Fig. 3a and Supplementary Fig. 2b). This further elaborates the benefits of carrying out the (ballistic) transport spectroscopy on understanding electronic properties of the system with SOC.

## Discussion

In summary, we have successfully demonstrated the ballistic electron motion in graphene on $WSe_2$ by measuring TMF signals at different

charge densities, magnetic fields, and temperatures. From the density and magnetic field dependence of the first focusing peaks (Fig. 2), we confirmed that there exist two split bands in the system as expected theoretically[2,4–7] and estimated the SOC strength of $\lambda_{SOC} = 13.0 \pm 4.7$ meV (Fig. 3a and Supplementary Fig. 2b). More interestingly, by analysing the behavior of the second focusing peak that shows no splitting and by carrying out quantum transport simulations, we were able to learn that the Rashba SOC is likely dominant in our system (Fig. 3b-f). Both the presence of the band splitting and a stronger Rashba SOC are well reproduced in SdH oscillations measurements (Fig. 5) even though they showed a smaller SOC strength of $\lambda_{SOC} = 3.4 \pm 0.7$ meV. A similar discrepancy was found in other 2DEG with Rashba SOC[19,21], indicating that there is a fundamental reason behind it. This calls for a new theory.

In addition to providing spectroscopic evidence of the spin-orbit-coupled bands, our work demonstrates that graphene on TMDCs can support ballistic transport that can be used not only to gain more insights into the microscopic electron process in the system but also to exploit various ballistic transport effects that are pronounced in graphene. It is particularly interesting as, in contrast to the existing studies on graphene spintronics[7,22,70,71], TMF separates spin-up and spin-down carriers in real space. This enables the detection and measurement of both spins independently, instead of only the majority one injected from magnetic contacts. This, therefore, offers an alternative venue for graphene spintronic applications[7,22,70,71]. Similar strategies can also be used to study other 2D materials or heterostructures with strong SOC, such as bilayer graphene-TMDC heterostructures[10–13], black phosphorus[72], and more, which will offer new understandings about the effect of SOC in these material systems.

## Methods

### Sample fabrication

The WSe$_2$, hBN, and graphene flakes were exfoliated from corresponding crystals onto silicon wafers and examined under an optical microscope. The flakes with suitable thicknesses and surfaces were selected and assembled onto highly doped silicon substrates with 285-nm-thick oxide, following the standard dry pick-up and transfer technique[46]. The WSe$_2$ flakes used in this study are around 20 nm and 40 nm for samples 1 and 2, respectively. After the assembly, the stacks were annealed at 250 °C for 2 h in a tube furnace in Ar/H$_2$ forming gas. 1D electrical contacts were fabricated on the annealed sample through a standard electron-beam lithography and reactive-ion etching (CF$_4$/O$_2$ mixture gas with flow rates of 5/25 sccm, RF power: 60 W), followed by electron beam evaporation of 5 nm Cr and 50 nm Au films. The devices were finally shaped into Hall bars by another electron-beam lithography and reactive-ion etching process.

### Electrical measurement

Devices were measured in a 1.5 K cryogen-free variable temperature insert (VTI) with a superconducting magnet. The electrical signals were measured by applying a small low-frequency (17.777 Hz) AC current of 0.1–1 μA between the source and drain terminals and measuring the voltage drop between another two probes using a lock-in amplifier (Stanford Research SR830). The low-noise filters and amplifiers were used to detect small TMF signals. The back gate was controlled by Keithley 2400 source-meter.

## Data availability

The data used in this study are freely available in the figshare database at https://doi.org/10.6084/m9.figshare.22644469.

## Code availability

Codes to analyze the data and perform numerical calculations are available upon reasonable request.

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

## Acknowledgements

The work is financially supported by the National Key R&D Program of China (2020YFA0309600) and by the University Grants Committee/

Research Grant Council of Hong Kong SAR under schemes of Area of Excellence (AoE/P-701/20), ECS (27300819), and GRF (17300020, 17300521, 17309722). K.W. and T.T. acknowledge support from the JSPS KAKENHI (Grant Numbers 21H05233 and 23H02052) and World Premier International Research Center Initiative (WPI), MEXT, Japan. N.W. acknowledges support from William Mong Institute of Nano Science and Technology. W.-H.K. and M.-H.L. gratefully acknowledge National Science and Technology Council of Taiwan (grant numbers: MOST 109-2112-M-006-020-MY3 and NSTC 112-2112-M-006-019-MY3) for financial support and National Center for High-performance Computing (NCHC) for providing computational and storage resources.

## Author contributions

D.-K.K. conceived and supervised the project. M.-H.L. supervised the theoretical part carried out by W.-H.K. Q.R. fabricated the samples and performed the measurements, assisted by H.X. Some of the data was collected in NW's group with help from Z.Y. and X.F. Q.R. and W.-H.K. analyzed the data, and D.-K.K. and M.-H.L. interpreted them with input from all authors. T.T. and K.W. synthesized the hBN crystals. Q.R., W.-H.K., M.-H.L., and D.-K.K. wrote the paper with input from all authors. All authors discussed the results.

## Competing interests

The authors declare they have no competing interests.
