## [Peer Review File · Nature Communications]

Reviewers' Comments:

Reviewer #1:

Remarks to the Author:

In the manuscript, Rao et. al. measure the effect of spin orbit coupling (SOC) on graphene proximitized by WSe_2 by carrying out transverse magnetic focusing (TMF) measurements. They observe clearly in ballistic orbits for which no bounces occur, there is a clear splitting of orbits. On subsequent TMF bounces, the peaks are no longer split. The authors go through careful analysis and conclude that this is primarily a consequence of Rashba SOC (as opposed to disordered scattering) wherein the orbits jump from one spin channel to the other. Beyond this core observation, the authors measured temperature dependence of the TMF to determine e-e scattering lengths, and measured SdH oscillations to perform a secondary probe of SOC. They observe SOC in the SdH oscillations, but interestingly at a lower magnitude.

To this reviewer, this work is carefully done, and shows that graphene proximitized by WSe_2 enables SOC in a clean electronic system. The work uses simulations to help understand features of the TMF data to bolster the central claim that the results indicate Rashba SOC. Largely, they are able to see similar phenomenology to those observed in 2DEGs, positioning proximitized graphene as a new material system to study/engineer SOC. As such, this work appears relevant and a good candidate for Nature Communications.

The only part that felt under-developed is the discussion of the $T^{-1.8}$ temperature dependence. Indeed, it is between T^{-1} and T^{-2} , but it seems it would be reasonable to put this in the context of existing work. Perhaps the screening of e-e electrons contributes to this power law, but there are also observations of roughly $T^{-1.8}$ dependence in works such as Nature Phys. 13, 1182 (2017). At least in the interpretation therein, this dependence is understood/expected without any considerations of a particularly strong dielectric. It would be good if the authors can attempt to make a more nuanced assessment of whether the WSe_2 is affecting the T-dependence or not.

With revisions associated with above discuss, this manuscript should be suitable for publication.

One other comment: the circular plots in Fig. 3a,b seem to have too much prominence for the amount of information that once can glean off of the figure. It seems that the authors are determining a radius and a confidence interval about that radius—This seems to be sufficient to be stated and not plotted in the main text.

Reviewer #2:

Remarks to the Author:

The study successfully demonstrated the ballistic electron motion in graphene on WSe_2 and confirmed the existence of split bands, with an estimated strength of SOC. Ballistic transverse magnetic focusing was used to observe an apparent splitting in the first peak, indicating a SOC strength of ~ 13 meV, and no splitting in the second peak, indicating stronger Rashba SOC. Temperature dependence measurements suggested the possible suppression of electron-electron scatterings. However, there was a discrepancy in the estimated SOC strength between the TMF and SdH oscillations measurements, indicating the need for a new theory. The work also highlighted the potential of graphene on TMDCs to exploit various ballistic transport effects and provide an alternative venue for graphene in spin-orbitronics. The paper is well-organized and will attract the attention of readers in emerging fields such as valleytronics, spin-orbitronics, and future heterogeneous electronics. Some minor revisions could be made for clarity, but overall the study is well-conducted, and the results are significant and interesting.

1. The authors focused on the hole side in their work on transverse magnetic focusing (TMF) due to the higher sample quality of their monolayer graphene on WSe_2 . While the band structure of 2D hole systems in GaAs is fundamentally different from equivalent electron systems, it is expected that the graphene case exhibits a similar TMF between hole and electron. To provide a reference for future works, it would be valuable to include TMF spectra on the electron side, even

if the splitting of the first peak is absent, in the Supplementary Information.

2. In the 2nd paragraph on page 6, the authors discussed the uneven heights of the two first focusing peaks. The spin polarization ratio in 2DEG was shown to be determined by the adiabatic transitions between two different spin subbands at the injector [Ref. 48]. The authors further estimated the strength of Rashba SOC as $\lambda_{\text{R}} > 0.25 \sim 0.35$ meV. However, it is not clear from the text how this estimation of λ_{R} relates to the uneven heights of the peaks. A more thorough explanation is needed to clarify the reason behind the uneven heights of the two first peaks.

3. The absence of splitting in the second TMF peak suggests that the Rashba spin-orbit coupling (SOC) is dominant in monolayer graphene on WSe_2 . As the gate voltage can modify the strength of the Rashba SOC, the splitting of the first peak is expected to increase with increasing gate voltage and carrier density. However, the TMF spectra shown in Fig. 2 and Fig. 5e exhibit only a slight increase in the splitting of the first peak with increasing carrier density. It would be helpful to provide an explanation for this observation.

4. In the 197th line on page 8, the authors wrote "Thus, the intervalley scattering can lead to the backscattering between the S_{\pm} bands at the edge when the spin-valley Zeeman term dominates, suppressing the splitting in the second peak." This statement is confusing. It is unclear whether the splitting in the second peak becomes stronger or weaker when the spin-valley Zeeman term dominates. Please clarify this point in the paper.

5. It is recommended to present $\Delta B/B$ in Fig. 5e since the magnitude of ΔB depends on the distance ($L = 2r_{\text{c}}$) between the injector and collector. This presentation will also have the benefit of facilitating a comparison with the data in the previous report [Ref. 21].

Reviewer #3:

Remarks to the Author:

Reviewer report to the manuscript NCOMMS-23-16631

The authors apply a magnetic focusing technique to collect those ballistic electrons coupled with strong spin-orbit coupling (SOC) in monolayer graphene on WSe_2 . The graphene, which benefits from its Dirac physics and dry-transfer fabrication, can provide a ballistic transport channel and WSe_2 , which can remain highly resistive so will not cause parallel conduction with graphene, can split the graphene bands by spin-orbit proximity effect. The spin-orbit split graphene bands can therefore be probed by studying the splitting of the first magnetic focusing peak as demonstrated by the authors. The details were further revealed by the higher-order focusing peaks, spin Hall effect, and Shubnikov-de Hass (SdH) oscillations. Finally, the results show that different charge or spin dynamics should be involved to explain the observed discrepancy of the SOC strength parameter from magnetic focusing and SdH oscillations.

I do not doubt the significance of ballistic transport spectroscopy for the currently studied graphene/ WSe_2 and other van der Waals heterostructure because it enables the direct on-chip detection of the unique band dispersion due to the complex interplay with the dielectric environment and interfaces. However, although the simulation results are beautiful, several of the analyses and methods for the experimental data and also the physics behind them have to be clarified and more clearly explained as shown by the point-to-point comment below.

1. The thickness of the WSe_2 layer has to be given because it can be a reference for its bandgap value estimation. It would be better that the transfer characteristic of WSe_2 as a transistor channel itself can be measured individually to check that there is no parallel conduction through WSe_2 , which here should be an insulating substrate for the spin-orbit proximity. I understand that the WSe_2 conducting channel can be pinched off over the whole gate voltage range by using the suitable source and drain contact metals [Avsar et al., Nat. Commun. 5, 4875 (2014)]. The authors should add more supplementary data or at least comment on and discuss with this issue.

2. In this work, the estimated carrier density rather than the directly applied gate voltage is used for the horizontal axis when studying the carrier density dependence of the data. How those densities were evaluated have to be described in more details. I speculate that they may come from capacitor modeling because the conventional four-terminal R_{xx} data in Fig. 1c are also plotted against the carrier density. Moreover, I also suggest that the simulation trace for the relation between the carrier density and magnetic field for each quantum Hall state of various filling factors should be included in support of the monolayer claim.

3. The mechanism for the important spin-orbit proximity effect needs more explanation. For example, whether its strength remains the same and how it changes with the operation gate voltage remain unclear. It has been shown that the SOC enhancement occurs only when the intrinsic defects in the substrate of strong SOC can act as a sink for conducting charges in the channel of weak SOC in the graphene/ WS_2 heterostructure (i.e., those defects in WS_2 create electronic states that overlap in energy with the electronic states of graphene) [Avsar et al., Nat. Commun. 5, 4875 (2014)]. According to this mechanism, SOC enhancement and then the claimed spin-orbit split bands can only occur above a characteristic threshold gate voltage. The present data do not reveal whether this mechanism works in the studied sample because of the observed weak dependence of the focusing peak splitting on the gate voltage.

4. The authors experimentally found that the height of the split peak closer to the zero magnetic field is always lower than that of the other within the first focusing peak doublet (line 145) and attribute it to the spin polarization. However, the simulation focusing spectra show opposite results (Figs. 1b and 3d). I can understand that the spontaneous spin polarization is not taken into account in their model. Nevertheless, the discrepancy between the experimental and theoretical peak heights has to be clarified.

5. The current manuscript does not provide the experimental methods for the individual control of spin-valley Zeeman and Rashba SOC terms (λ and λ_2 , respectively), which is essential for obtaining the average of the normalized difference between the focusing data and the simulation $\langle \delta B \rangle$ as a function of λ and λR (Figs. 3a, 3b, and 5d). The impact of this work would be significantly raised if these two distinct spin-related mechanisms can indeed be individually adjusted.

6. It seems that a crossover from diffusive to ballistic charge transport and invisible to visible focusing peaks occur with the gate voltage (Fig. 2a). Accordingly, the extrinsic spin Hall effect should be the origin for the observed large nonlocal resistance near zero carrier density (Fig. 2a). Further discussion is necessary to comprehend the physics behind the potential coexistence of diffusive spin Hall effect and ballistic transverse magnetic focusing.

To all reviewers

We sincerely appreciate the time and efforts of the reviewers to carefully read our manuscript and help us to make it more complete. Thanks to their comments, we have realized that some points in the original manuscript had not been clearly discussed and presented, which may have led to some misunderstanding in a few specific aspects. We have modified the manuscript and supplementary information to address all the points raised by the referees. We are grateful to the referees because, as a result, our manuscript has become stronger.

The point-by-point reply to the comments of the referees follows here below.

Reviewer #1

Comment: *In the manuscript, Rao et. al. measure the effect of spin orbit coupling (SOC) on graphene proximitized by WSe₂ by carrying out transverse magnetic focusing (TMF) measurements. They observe clearly in ballistic orbits for which no bounces occur, there is a clear splitting of orbits. On subsequent TMF bounces, the peaks are no longer split. The authors go through careful analysis and conclude that this is primarily a consequence of Rashba SOC (as opposed to disordered scattering) wherein the orbits jump from one spin channel to the other. Beyond this core observation, the authors measured temperature dependence of the TMF to determine e-e scattering lengths, and measured SdH oscillations to perform a secondary probe of SOC. They observe SOC in the SdH oscillations, but interestingly at a lower magnitude.*

To this reviewer, this work is carefully done, and shows that graphene proximitized by WSe₂ enables SOC in a clean electronic system. The work uses simulations to help understand features of the TMF data to bolster the central claim that the results indicate Rashba SOC. Largely, they are able to see similar phenomenology to those observed in 2DEGs, positioning proximitized graphene as a new material system to study/engineer SOC. As such, this work appears relevant and a good candidate for Nature Communications.

Reply: We are glad to read that the referee appreciates the context and the relevance of our work. In particular, we are pleased to read that the referee considers our work “carefully done” and “positioning proximitized graphene as a new material system to study/engineer SOC”. The latter is indeed the major claim that we would like to deliver in this paper.

Comment 1: *The only part that felt under-developed is the discussion of the T-1.8 temperature dependence. Indeed, it is between T-1 and T-2, but it seems it would be reasonable to put this in the context of existing work. Perhaps the screening of e-e electrons contributes to this power law, but there are also observations of roughly T-1.8 dependence in works such as Nature Phys. 13, 1182 (2017). At least in the interpretation therein, this dependence is understood/expected without any considerations of a particularly strong dielectric. It would be good if the authors can attempt to make a more nuanced assessment of whether the WSe₂ is affecting the T-dependence or not.*

With revisions associated with above discuss, this manuscript should be suitable for publication.

Reply 1: We thank the referee for bringing up this important issue about the temperature dependence. Let us first elaborate on our logic behind the reasoning to take the WSe₂ as a source of such a weak temperature dependence. While analyzing the data, we spotted this weak temperature dependence but could not find a theory to explain the phenomenon. Thus, we have simply compared the data with similar

TMF studies in graphene-hBN heterostructures (Refs. 24 and 25 of the revised manuscript), where they found $1/T^2$ dependence in a wide range of density and temperatures. Since the most obvious difference between these studies and ours is the presence of WSe_2 and SOC, we suspected that it is either the WSe_2 or SOC, or both, that affected the temperature dependence. As we were unclear about how the SOC can affect electron-electron interaction and found that the WSe_2 has a dielectric constant larger than the hBN, we used the dielectric screening as a possible origin for the weak temperature dependence without providing more supporting arguments.

During this analysis, we compared our data with only TMF studies to keep other variables the same. For instance, the paper [Nat. Phys. 13, 1182 (2017)] the referee suggested and Ref. 61 in the revised manuscript have measured transport through a constriction and extracted contributions from a viscous flow to get electron-electron scattering length. Although their analysis is interesting and can be relevant to our study, we find that these studies “extracted” electron-electron scattering lengths, so it seems that they did not look at the total electron scattering length as in our case. Also, we found that in the 2017 Nat. Phys. paper, the deviation from $1/T^2$ dependence occurs in 50-100 K when the temperature is not insignificant with respect to the Fermi energy, which can be understood in terms of electron-electron interaction in conventional Fermi liquids. In contrast, our data show the $1/T^{1.8}$ dependence from ~ 5 K for different charge densities. Thus, we believe the direct comparison between our work and the previous works on the extracted electron-electron scattering length may not be appropriate.

However, while trying to find other possible reasons for the weak temperature dependence, as the referee suggested, we found several recent works (Refs. 59 and 60 of the revised manuscript) that claim that the SOC induced by WSe_2 can influence electronic interactions such as superconductivity in graphene systems. This might suggest that the induced SOC can also be a reason why we have a weaker temperature dependence in TMF, although the origin is still unclear. As it can be interesting for future studies, we have revised the relevant text on page 11 as follows:

“... electron-electron ($e-e$) scattering (T^{-2}), indicating that although the $e-e$ scattering is dominant, it is also slightly suppressed in our sample. In comparison, similar TMF studies on graphene-hBN heterostructures^{24,25} have shown T^{-2} dependence in a wide range of temperatures and charge density. Thus, the $T^{-1.8}$ dependence found in our sample should be from the WSe_2 itself and/or the induced SOC. Although the exact origin is unclear, we note that the WSe_2 has a dielectric constant ($\epsilon_0 \approx 7.9$) about twice larger than h-BN used in the previous studies ($\epsilon_0 \approx 3.8$)⁵⁸ that may induce more screening. In addition, recent studies^{59,60} have shown that the SOC can affect electronic interaction phenomena such as superconductivity in twisted or Bernal bilayer graphene. This suggests a possibility to measure ballistic transport effects in graphene-TMDC heterostructures to study ...”

Comment 2: *One other comment: the circular plots in Fig. 3a,b seem to have too much prominence for the amount of information that once can glean off of the figure. It seems that the authors are determining a radius and a confidence interval about that radius—This seems to be sufficient to be stated and not plotted in the main text.*

Reply 2: The referee is correct. In Figs. 3a,b of the original manuscript, the radius and the width of the donut are the two key parameters, and the figures might stand out too much more than the information that they deliver, but the figures are also used to define the meanings of the θ_{SOC} and $\langle \delta B^2 \rangle$ which is used in other parts of the paper. Thus, we find it better to keep at least one of them (Fig. 3a of the revised manuscript) and move another (Fig. 3b) to the supplementary (Supplementary Fig. 2b). We have also changed the color scale of Fig. 3a to match with other figures in Fig. 3.

Reviewer #2

Comment: The study successfully demonstrated the ballistic electron motion in graphene on WSe₂ and confirmed the existence of split bands, with an estimated strength of SOC. Ballistic transverse magnetic focusing was used to observe an apparent splitting in the first peak, indicating a SOC strength of ~13 meV, and no splitting in the second peak, indicating stronger Rashba SOC. Temperature dependence measurements suggested the possible suppression of electron-electron scatterings. However, there was a discrepancy in the estimated SOC strength between the TMF and SdH oscillations measurements, indicating the need for a new theory. The work also highlighted the potential of graphene on TMDCs to exploit various ballistic transport effects and provide an alternative venue for graphene in spin-orbitronics. The paper is well-organized and will attract the attention of readers in emerging fields such as valleytronics, spin-orbitronics, and future heterogeneous electronics. Some minor revisions could be made for clarity, but overall the study is well-conducted, and the results are significant and interesting.

Reply: We are pleased to read that the referee acknowledges that our paper is “well-organized” and “will attract the attention of readers in emerging fields” and that “the study is well-conducted, and the results are significant and interesting.” We also thank the referee for making comments that helped us to improve our manuscript, as shown in our point-by-point reply below.

Comment 1: 1. The authors focused on the hole side in their work on transverse magnetic focusing (TMF) due to the higher sample quality of their monolayer graphene on WSe₂. While the band structure of 2D hole systems in GaAs is fundamentally different from equivalent electron systems, it is expected that the graphene case exhibits a similar TMF between hole and electron. To provide a reference for future works, it would be valuable to include TMF spectra on the electron side, even if the splitting of the first peak is absent, in the Supplementary Information.

Reply 1: We fully agree with the referee that it is important to share the data in all ranges of parameters for future studies. We have therefore included the 1D cuts of the data from sample 1 on the electron side in the supplementary (Supplementary Fig. 2a). For easy access, we have copied the figure below (Fig. R1).

Figure R1. 1D cuts of the data from sample 1 on the electron side shown in Fig. 2a. The figure is identical to Supplementary Fig. 2a.

Comment 2: 2. In the 2nd paragraph on page 6, the authors discussed the uneven heights of the two first focusing peaks. The spin polarization ratio in 2DEG was shown to be determined by the adiabatic

transitions between two different spin subbands at the injector [Ref. 48]. The authors further estimated the strength of Rashba SOC as $\lambda_R > 0.25 \sim 0.35$ meV. However, it is not clear from the text how this estimation of λ_R relates to the uneven heights of the peaks. A more thorough explanation is needed to clarify the reason behind the uneven heights of the two first peaks.

Reply 2: We thank the referee to raise this point as while preparing the reply, we have found that there must be more rigorous analysis and discussions when one wants to identify the possible origin of the spin polarization that leads to the uneven heights of the peaks.

Before discussing the details, we would first like to give a direct answer to the referee's question. Basically, we have followed the discussions in Ref. 51 of the revised manuscript, where it was argued that the spin polarization can occur when the current is injected through the constriction with the width W_0 , which can give rise to the uneven heights of the TMF peaks as we found in our experiments. The condition for this is that $k_\alpha/k_F > (3/16)(\lambda_F/W_0)^2$ when W_0 , k_F and λ_F are the width of the constriction at the injection point, the Fermi wavenumber and wavelength, respectively, and k_α is the half of the splitting of the bands in momentum space due to the Rashba SOC. From the definition of the parameters and comparing them with the Hamiltonian of our system, Eq. (1) of the manuscript, we could derive the condition for spin polarization in our sample $\frac{\Delta k}{2k_F} > \frac{3}{16} \left(\frac{\lambda_F}{w}\right)^2$ as shown in the manuscript. From this, we estimated $\lambda_R > \frac{3\pi}{8} \frac{\hbar v_F \lambda_F}{w^2} \approx 0.25 \sim 0.35$ meV.

While looking for more solid evidence to connect the uneven heights of the TMF peaks to the origin of the spin polarization, we have realized that it requires more rigorous analysis and controlled experiments. First, we should note that the argument above assumes the formation of the sub-bands at the contact, which cannot be confirmed experimentally in our device as there is no quantum point contact that can be used to tune the width W_0 to detect clear conductance quantization. Second, the recent study by Rendell et al. [PRB 107, 045304 (2023); Ref. 50 of the revised manuscript] showed that the heights of the split-peaks depend on the distance between the injector and collector, d , as shown in Fig. R2 below (figure 4 of the paper).

Figure R2. This is the figure 4 taken from PRB 107, 045304 (2023) that shows TMF signal as a function of magnetic field for the samples with different values of d . It clearly shows that the relative heights of the HH- and HH+ peaks vary with the distance d .

This is from the difference in the scattering rate for the electrons at different spin-orbit-split bands, which needs to be considered for quantitative analysis. In addition, the study on 2D hole gas [PRL 106, 236601 (2011); Ref. 49 of the revised manuscript] showed that the cubic Rashba SOC term in the system generates anomalous spin polarization. Since our system has two SOC terms, it can be important to consider the effect of these two terms on TMF if we want to discuss spin polarization from the uneven heights of the focusing peaks. It is therefore not straightforward to discuss the origin of spin polarization from the uneven heights of the focusing peak in our study.

However, we emphasize that all these studies have indeed used the uneven heights of the focusing peaks as a “signature” of the spin polarization in the system as well as an experimental parameter to quantify the spin polarization as a function of other experimental parameters such as in-plane magnetic fields, even though more rigorous studies were needed to “identify” the origin of such spin polarizations. Therefore, we believe our major claim that “our study shows a possibility of using TMF to detect spin polarization of the ballistic carriers in graphene-TMDC heterostructures” (page 8 in the revised manuscript) still holds. To make the discussions more clear and complete, we have modified the relevant paragraph as follows:

Page 7 – “... of the second (see, e.g., **Figs. 2c,d**). Interestingly, in 2DEG with Rashba SOC, such uneven heights of the split peaks have been used as a signature of the spin polarization in the system, even though more rigorous analysis and more controlled experiments are needed to identify the origin of the spin polarization^{17,48-51}. For instance, it was suggested that adiabatic transition between the quantized sub-bands formed at the injector with width w could polarize...”

Pages 7 and 8 – “... well in the range (**Fig. 3a**), but due to the absence of the quantum point contacts in our device, we cannot confirm the formation of the sub-bands at the injector. On the other hand, it was also shown that the relative heights of the split peaks could vary with the distance between the injector and detector due to the difference in scattering lengths for carriers at different spin-orbit-split bands⁵⁰ and that the exact momentum dependence of the SOC should be considered to understand the spin-polarization⁴⁹. Interestingly, in graphene on TMDCs, the presence of both spin-valley Zeeman and Rashba SOC terms was predicted to induce a characteristic spin winding of the spin-orbit-coupled bands that leads to a current-induced spin polarization⁵², which may result in the uneven heights of the peaks. For better understanding, we will need to conduct more sophisticated experiments, such as applying in-plane magnetic fields to control Zeeman energy^{17,49,53}, using samples with various distances between the injector and detector⁵⁰, or using ferromagnetic contacts for spin-sensitive detection⁵⁴. Nevertheless, we note that all these previous studies have used the uneven heights of the split peaks as a signature of the spin polarization in the system^{17,48-51}. Thus, our study shows a possibility of using TMF to detect spin polarization of the ballistic carriers in graphene-TMDC heterostructures.”

Comment 3: *3. The absence of splitting in the second TMF peak suggests that the Rashba spin-orbit coupling (SOC) is dominant in monolayer graphene on WSe₂. As the gate voltage can modify the strength of the Rashba SOC, the splitting of the first peak is expected to increase with increasing gate voltage and carrier density. However, the TMF spectra shown in Fig. 2 and Fig. 5e exhibit only a slight increase in the splitting of the first peak with increasing carrier density. It would be helpful to provide an explanation for this observation.*

Reply 3: As the referee wrote, Rashba SOC strength is known to be tuned by the perpendicular electric field. However, in our device, we have only one back gate, and moreover, the WSe₂ will become conducting at too large back gate voltage such that the maximum possible electric field that we could

apply should be around 0.13 V/nm which is too small to make a sizable change of Rashba SOC strength in our samples.

For reference, we found one study that measured spin-relaxation time as a function of the electric field in graphene-TMDC heterostructures using both top and bottom gates [2D Mater. 3, 031012 (2016)]. In the study, the spin relaxation rate (τ_R^{-1}) was shown to increase from around 0.2 ps⁻¹ to 0.24 ps⁻¹ at the electric field changing from -0.2 V/nm to 0.2 V/nm when the momentum scattering rate (τ_e^{-1}) is fixed at 12 ps⁻¹. Since the study claimed that the Dyakonov-Perel mechanism governs spin relaxation in the system, we could calculate the change of Rashba SOC strength from the relation $\tau_R^{-1} = 2\tau_e \left(\frac{\lambda_R}{\hbar}\right)^2$ that shows the change from ~0.72 meV to ~0.79 meV. This is only around an 8.8% change which is already within the error in our fitting (see Fig. 3a and related discussion) even before considering that in our case, the maximum possible electric field is only around 0.13 V/nm.

To be clear, we must note that to our knowledge, all other studies, except for the 2016 2D Materials paper above, have not shown clear signatures of the change of Rashba SOC strength as a function of the electric field in monolayer graphene-TMDC heterostructures. In addition, the 2016 study above mainly used weak anti-localization measurement to extract the spin-relaxation time that can give large sample-to-sample variations as shown in Fig. 5d of our manuscript. Therefore, at this moment, we believe there is no clear experimental evidence for the electric-field-tuned SOC in monolayer graphene-TMDC heterostructures. This can be related to the fact that the electron wave in monolayer graphene is confined to only one atom-thin layer which could significantly reduce the effect of the perpendicular electric field.

Comment 4: *4. In the 197th line on page 8, the authors wrote “Thus, the intervalley scattering can lead to the backscattering between the S_{\pm} bands at the edge when the spin-valley Zeeman term dominates, suppressing the splitting in the second peak.” This statement is confusing. It is unclear whether the splitting in the second peak becomes stronger or weaker when the spin-valley Zeeman term dominates. Please clarify this point in the paper.*

Reply 4: The complication comes from the fact that the Rashba and spin-valley Zeeman terms create different spin-winding for different valleys. For this, we have extended Figs. 3b-e (Fig. 3c in the original manuscript) to show the spin orientations of the bands at both valleys (see the figure at the next page for easy reference). As shown in the revised figure, the spin winding for S_{\pm} bands remains the same for different valleys when only the Rashba term exists (Figs. 3d,e) but it becomes the opposite when the spin-valley Zeeman term is dominant (Figs. 3b,c). Therefore, the intervalley scattering at the edge can turn the electron at the state A at K valley to C' in K' valley, i.e. the backscattering between the S_{\pm} band, when the spin-valley Zeeman term dominates.

In addition to revising the figure, we have also modified the text as follows:

Page 8 – “... bands S_{\pm} . For more accessible discussion, let us first consider a single valley only and discuss the effect of the intervalley scattering later. As depicted ...”

Page 9 – “... different valleys (see **Fig. 3c**). Thus, the intervalley scattering can lead to the scattering from state A at K valley to state C' at K' valley, i.e. the backscattering between the S_{\pm} bands at the edge, when the spin-valley Zeeman term dominates, suppressing ...”

Figure R3. The figure is copied from **Figs. 3b-e**.

Comment 5: 5. It is recommended to present $\Delta B/B$ in Fig. 5e since the magnitude of ΔB depends on the distance ($L = 2rc$) between the injector and collector. This presentation will also have the benefit of facilitating a comparison with the data in the previous report [Ref. 21].

Reply 5: We thank the referee for raising this point. Initially, we used ΔB as it is directly related to the $k_{F1} - k_{F2}$ that seemed simpler to compare the data from TMF and SdH measurements, whereas the $\Delta B/B$ has a denominator that varies with $(k_{F1} + k_{F2})/2$. However, we noticed that $\Delta B/B$ has only momentum dependence such that it can be used to compare the results from other studies such as Ref. 21 of the revised manuscript. We have changed the figure (Fig. 5e in the revised manuscript; see below) and revised relevant discussions on page 12 as follows.

Figure R4. The normalized splitting $\Delta B/B$ of the first TMF peak at different charge densities extracted from **Fig. 2** compared with the splitting calculated from the SdH oscillations.

Page 12 – “... together in **Fig. 5e** after normalizing the values with the averaged peak positions of the two sub-peaks. As shown in the figure, $\Delta B_{TMF}/B_{TMF}$ remains larger than $\Delta B_{SdH}/B_{SdH}$ in all density ...”

Reviewer #3

Comment: The authors apply a magnetic focusing technique to collect those ballistic electrons coupled with strong spin-orbit coupling (SOC) in monolayer graphene on WSe₂. The graphene, which benefits from its Dirac physics and dry-transfer fabrication, can provide a ballistic transport channel and WSe₂, which can remain highly resistive so will not cause parallel conduction with graphene, can split the graphene bands by spin-orbit proximity effect. The spin-orbit split graphene bands can therefore be probed by studying the splitting of the first magnetic focusing peak as demonstrated by the authors. The details were further revealed by the higher-order focusing peaks, spin Hall effect, and Shubnikov-de Hass (SdH) oscillations. Finally, the results show that different charge or spin dynamics should be involved to explain the observed discrepancy of the SOC strength parameter from magnetic focusing and SdH oscillations.

I do not doubt the significance of ballistic transport spectroscopy for the currently studied graphene/WSe₂ and other van der Waals heterostructure because it enables the direct on-chip detection of the unique band dispersion due to the complex interplay with the dielectric environment and interfaces. However, although the simulation results are beautiful, several of the analyses and methods for the experimental data and also the physics behind them have to be clarified and more clearly explained as shown by the point-to-point comment below.

Reply: We are pleased to read that the referee has no doubt about “the significance of ballistic transport spectroscopy for the currently studied graphene/WSe₂” which is a major claim of our work. We would also like to thank the referee for raising several key comments that allow us to improve our manuscript.

Comment 1: 1. The thickness of the WSe₂ layer has to be given because it can be a reference for its bandgap value estimation. It would be better that the transfer characteristic of WSe₂ as a transistor channel itself can be measured individually to check that there is no parallel conduction through WSe₂, which here should be an insulating substrate for the spin-orbit proximity. I understand that the WSe₂ conducting channel can be pinched off over the whole gate voltage range by using the suitable source and drain contact metals [Avsar et al., Nat. Commun. 5, 4875 (2014)]. The authors should add more supplementary data or at least comment on and discuss with this issue.

Reply 1: The referee is correct. It has been known that at large gate voltages, the TMDC substrate can become conducting which pinches off the transport through graphene. To clarify this, we have added the thicknesses of the WSe₂ used in devices 1 and 2 in the methods section and plot the conductance as a function of the gate voltage in Supplementary Fig. 1a (copied below).

Figure R5. The four-terminal conductance G as a function of the carrier density n . Taken from Supplementary Fig. 1a.

As shown in the figure, the WSe₂ flake for sample 1 remains insulating over the carrier density range where we measured TMF signals, while WSe₂ for sample 2 becomes conducting at around $n \gtrsim 1 \times 10^{12} \text{ cm}^{-2}$. We would, however, like to emphasize that the range of the gate voltage, when the pinch-off happens, depends on the type of the TMDC and the sample quality. In general, we found that the WSe₂ provides a larger range of gate voltage in which one can study graphene transport, compared with WS₂ or MoS₂ as shown in the figure below taken from the Fig. 1 of Ref. 3. In fact, this is one of the reasons why we used WSe₂ to access a larger charge density range for ballistic transport studies.

Figure R6. Taken from PRX 6, 041020 (2016) which is the Ref. 3 of the manuscript. We can clearly see that above certain gate voltages, the conductance of graphene doesn't change because the TMDC becomes conducting.

Comment 2: 2. In this work, the estimated carrier density rather than the directly applied gate voltage is used for the horizontal axis when studying the carrier density dependence of the data. How those densities were evaluated have to be described in more details. I speculate that they may come from capacitor modeling because the conventional four-terminal R_{xx} data in Fig. 1c are also plotted against the carrier density. Moreover, I also suggest that the simulation trace for the relation between the carrier density and magnetic field for each quantum Hall state of various filling factors should be included in support of the monolayer claim.

Reply 2: We thank the referee for their suggestions. We have measured the classical Hall effect at a small magnetic field at different gate voltages and used the data to estimate the capacitance, which is 0.66×10^{11} and $1.56 \times 10^{11} \text{ cm}^{-2}/\text{V}$ for devices 1 and 2. The figure on the left shows the Hall density as a function of the gate voltage for device 1 as an example whereas the right figure shows the map of R_{xx} as a function of the filling factor and magnetic field that confirms the monolayer graphene. These figures are included in the supplementary.

Figure R7. Taken from Supplementary Fig. 1

Comment 3: 3. The mechanism for the important spin-orbit proximity effect needs more explanation. For example, whether its strength remains the same and how it changes with the operation gate voltage remain unclear. It has been shown that the SOC enhancement occurs only when the intrinsic defects in the

substrate of strong SOC can act as a sink for conducting charges in the channel of weak SOC in the graphene/WS2 heterostructure (i.e., those defects in WS2 create electronic states that overlap in energy with the electronic states of graphene) [Avsar et al., Nat. Commun. 5, 4875 (2014)]. According to this mechanism, SOC enhancement and then the claimed spin-orbit split bands can only occur above a characteristic threshold gate voltage. The present data do not reveal whether this mechanism works in the studied sample because of the observed weak dependence of the focusing peak splitting on the gate voltage.

Reply 3: As the referee wrote, the study by Avsar et al (Ref. 1 in our manuscript) had claimed that the SOC is induced through defects and it occurs only above a certain threshold gate voltage. We would, however, like to clarify that in nearly all follow-up studies (Refs. 2,3,10-14, 16, and 38 of our manuscript to name a few), the signatures of the strong SOC have been found in all accessible density range down to zero. For instance, in the quantum capacitance measurement (Ref. 11 of the manuscript), it was found that the proximity-induced SOC in bilayer-TMDC heterostructures can open a gap at charge neutrality that indicates the strong effect of SOC even at zero density. Moreover, quantum transport measurements in Refs. 2-3 showed clear weak anti-localization signatures from strong SOC in a wide range of density. Further, in the SdH oscillation measurements in the monolayer (our work and Ref. 38) and in the bilayer (Ref. 3), the signature of the SOC-induced band splitting was found down to a very low density well within the gap of the TMDC. These works show that in graphene-TMDC heterostructures, the SOC is induced in all density ranges not just above a certain critical voltage, and that the mechanism is not defect related.

This is possible because the SOC in these devices, including ours, is induced through the proximity effect by the atomic potentials generated by the transition metals arranged in a TMDC layer. Such an atomic potential can influence the band structure of graphene quite effectively due to the atomically clean interface between the graphene and TMDC. Note that it is not necessary to fill the atomic orbitals of the transition metals for this. One prominent example is the graphene-hBN moiré heterostructures, where the different atomic potentials of the boron and nitrogen atoms combined with the moiré structure give rise to the cloning of the charge neutrality point in graphene band well within the energy gap of the hBN [see, for instance, Nat. Phys. 8, 382 (2012), Nature 497, 594 (2013), Nature 497, 598 (2013), and Science 340, 1427 (2013)]. We note that such an intrinsic proximity effect occurs in nearly all vdW heterostructures for all density ranges even without defects. In fact, our study clearly shows the signature of the SOC-induced band splitting in ballistic transport regime, which is not possible if the SOC is induced by defects. Thus, our work further supports the proximity-induced SOC in graphene-TMDC heterostructures without involving defects.

Comment 4: *4. The authors experimentally found that the height of the split peak closer to the zero magnetic field is always lower than that of the other within the first focusing peak doublet (line 145) and attribute it to the spin polarization. However, the simulation focusing spectra show opposite results (Figs. 1b and 3d). I can understand that the spontaneous spin polarization is not taken into account in their model. Nevertheless, the discrepancy between the experimental and theoretical peak heights has to be clarified.*

Reply 4: We thank the referee for pointing this out. Unfortunately, those curves (in Figs. 1d and 3f of the revised manuscript; Figs 1b and 3d of the original) from transport simulations are the so-called two-point conductance G between the injector and detector probes, i.e., $G = T(e^2/h)$ where T is the transmission function from the injector to the detector (or vice versa), according to the Landauer formalism. On the other hand, the experimentally measured TMF signal is based on the so-called four-point resistance (such as Fig. 2), which can also be simulated using the Büttiker formula. Computationally, however, the Büttiker

formula requires computing $N(N - 1)$ transmission functions between any pair of two different leads for an N -terminal system. Take our 10-terminal Hall bar (Fig. 1b) for example. There will be 90 transmission functions to compute if one really wishes to apply the Büttiker formula to compute the four-point resistance. Apparently, this is far beyond what our computation resources can afford. Therefore, comparing the trend of the peak heights of G from our simulations and our nonlocal resistance R_{nl} from our four-point measurement is unfortunately not appropriate. Nevertheless, the TMF peaks of the computed G growing with the weak magnetic field B can be simply understood as the enhanced transmission of the injected electrons. This is reasonable because, within the weak magnetic field regime, Bloch electrons behave rather semiclassically. The stronger the magnetic field, the more electrons are bent, and the more electrons are collected in the detector probe, leading to enhanced transmission from the injector to the detector.

To make this point clearer, we have added the relevant discussions at the end of the Supplementary Note 1 and specified “two-point conductance” in the caption of Fig. 1d as follows:

Supplementary Note 1 – “To simulate our TMF experiment done on the multi-terminal graphene Hall bar (Fig. 1b of the main text), considering the geometry as close to the real device as possible while keeping the computation affordable, we calculate the so-called two-point conductance G (those reported in Figs. 1d and 3f) between the injector and detector, i.e., $G = T(e^2/h)$ where T is the transmission function from the injector to the detector (or vice versa), based on the Landauer formula¹³. On the other hand, the experimentally measured TMF signal is based on the so-called four-point resistance (such as Fig. 2 of the main text), which can also be simulated using the Büttiker formula¹³. Computationally, however, the Büttiker formula requires computing $N(N - 1)$ transmission functions between any pair of two different leads for an N -terminal system.

In our case (see Fig. 1b of the main text), there are $10 \times 9 = 90$ transmission functions to compute, which is beyond our computation limit, even if we could afford to model the full size of our experimental device. In fact, instead of modeling the full device of our experiment, we used an effective three-terminal device of smaller area and shorter probe spacing ($L = 1.0 \mu\text{m}$) for our transport simulations, in order to lower the computation burden. Nevertheless, our two-point conductance calculations (for the effective three-terminal Hall bar) reveal consistent behaviors of the TMF peaks, compared to those from our four-point resistance measurement, including the splitting of the first peaks.”

Caption of Fig. 1d – “... **d** The corresponding TMF spectra, the two-point conductance G as a function of magnetic field, calculated at 6 different hole densities, n (in 10^{12} cm^{-2}) = -0.78, -0.93, -1.09, -1.24, -1.40, and -1.56 (from bottom to top) using the tight-binding model.”

While further searching for the possibility of using simulation to identify the origin of the spin polarization, we found that even in the experiment, it requires more rigorous work. First, the recent study by Rendell et al. [PRB 107, 045304 (2023); Ref. 50 of the revised manuscript] showed that the heights of the split peaks depend on the distance between the injector and collector, d , as shown in Fig. R8 below (figure 4 of the paper). This is from the difference in scattering rate for the electrons at different spin-orbit-coupled bands, which need to be considered or measured for quantitative analysis. In addition, the study on 2D hole gas [PRL 106, 236601 (2011); Ref. 49 of the revised manuscript] showed that the cubic Rashba SOC term in the system generates anomalous spin polarization. Since our system has two SOC terms, it can be important to consider the effect of these two terms on TMF if we want to discuss spin polarization from

the uneven heights of the focusing peaks. It is, therefore, not straightforward to discuss the origin of spin polarization from the uneven heights of the focusing peak.

Figure R8. This is the figure 4 taken from PRB 107, 045304 (2023) that shows TMF signal as a function of magnetic field for the samples with different values of d . It clearly shows that the relative heights of the HH- and HH+ peaks vary with the distance d .

However, we note that all these studies have used the uneven heights of the focusing peaks as the “signatures” of the spin polarization in the system and “experimentally measured parameters” to quantify the spin polarization as a function of other experimental parameters such as in-plane magnetic fields. Therefore, we believe that our major claim that “our study shows a possibility of using TMF to detect spin polarization of the ballistic carriers” (page 8 in the revised manuscript) still holds.

To make the discussions more rigorous, we have modified the relevant paragraph as follows:

Page 7 – “... of the second (see, e.g., **Figs. 2c,d**). Interestingly, in 2DEG with Rashba SOC, such uneven heights of the split peaks have been used as a signature of the spin polarization in the system, even though more rigorous analysis and more controlled experiments are needed to identify the origin of the spin polarization^{17,48-51}. For instance, it was suggested that adiabatic transition between the quantized sub-bands formed at the injector with width w could polarize...”

Pages 7 and 8 – “... well in the range (**Fig. 3a**), but due to the absence of the quantum point contacts in our device, we cannot confirm the formation of the sub-bands at the injector. On the other hand, it was also shown that the relative heights of the split peaks could vary with the distance between the injector and detector due to the difference in scattering lengths for carriers at different spin-orbit-split bands⁵⁰ and that the exact momentum dependence of the SOC should be considered to understand the spin-polarization⁴⁹. Interestingly, in graphene on TMDCs, the presence of both spin-valley Zeeman and Rashba SOC terms was predicted to induce a characteristic spin winding of the spin-orbit-coupled bands that leads to a current-induced spin polarization⁵², which may result in the uneven heights of the peaks. For better understanding, we will need to conduct more sophisticated experiments, such as applying in-plane magnetic fields to control Zeeman energy^{17,49,53}, using samples with various distances between the injector and detector⁵⁰, or using ferromagnetic contacts for spin-sensitive detection⁵⁴. Nevertheless, we

note that all these previous studies have used the uneven heights of the split peaks as a signature of the spin polarization in the system^{17,48-51}. Thus, our study shows a possibility of using TMF to detect spin polarization of the ballistic carriers in graphene-TMDC heterostructures.”

Comment 5: *5. The current manuscript does not provide the experimental methods for the individual control of spin-valley Zeeman and Rashba SOC terms (λ and λ_2 , respectively), which is essential for obtaining the average of the normalized difference between the focusing data and the simulation $\langle\delta B_2\rangle$ as a function of λ and λ_R (Figs. 3a, 3b, and 5d). The impact of this work would be significantly raised if these two distinct spin-related mechanisms can indeed be individually adjusted.*

Reply 5: We would first like to clarify that Figs. 3a and 5b, and Supplementary Fig. 2b of the revised manuscript (Figs. 3a, 3b, and 5d in the original) compare experimental data with theory, and the strengths of the SOC terms are used as fitting parameters. In practice, we first calculated the band structure by putting different values of the λ and λ_R in Eq. (1) from which we estimated the positions of the TMF peaks that are compared with the experimental data. It is therefore not needed to tune λ and λ_R experimentally to draw Figs. 3a and 5b, and Supplementary Fig. 2b.

However, we fully agree with the referee that if we could find or show a way to tune the spin-valley Zeeman and Rashba SOC terms independently, the impact of our work will become more significant. Although it is not controlled, we were at least able to show a stronger Rashba SOC in our device by analyzing the second TMF peak (Fig. 3) and SdH oscillations (Fig. 5) which have not been done in previous experiments. We believe it is one of the impacts of our work. As a future study, we are indeed planning to change the device geometry to tune the two SOC terms separately. For instance, we believe that in nearly perfectly aligned TMDC-graphene-TMDC heterostructures, the mirror symmetry will be preserved, suppressing the Rashba term. This would lead to a stronger spin-valley Zeeman term that can be observed as the splitting in the second focusing peak.

Comment 6: *6. It seems that a crossover from diffusive to ballistic charge transport and invisible to visible focusing peaks occur with the gate voltage (Fig. 2a). Accordingly, the extrinsic spin Hall effect should be the origin for the observed large nonlocal resistance near zero carrier density (Fig. 2a). Further discussion is necessary to comprehend the physics behind the potential coexistence of diffusive spin Hall effect and ballistic transverse magnetic focusing.*

Reply 6: We thank the referee to draw our attention to the crossover regime. As the referee wrote, Figs. 2a,b show that the TMF spectra nearly disappear in density below around $2.0\sim 3.0\times 10^{11}$ cm⁻² (see Supplementary Fig. 4 and Fig. R9 below). Interestingly, we found that it coincides well with the density below which the non-local signal becomes positive, i.e. the charge transport becomes diffusive (see the inset of Fig. 1e of the revised manuscript) and that the non-local signal becomes larger than the Ohmic contribution, i.e. the extrinsic SHE appears (Fig. 4b). The results, therefore, show that there is a crossover between the diffusive spin-Hall to ballistic TMF transport regime in our device at finite charge density, not the coexistence of these two effects at least within the resolution of our experiment.

Since it is another interesting point of studying ballistic transport in spin-orbit-coupled systems, we have added 1D cuts of TMF spectra near zero density in the supplementary information (Supplementary Fig. 4) to make the crossover more visible and briefly discuss this in the revised manuscript at the page 10 as follows:

“... in a similar system¹ previously, except that the signal appears near zero density below $2.0\sim 3.0 \times 10^{11} \text{ cm}^{-2}$ in our device. We attribute this to the crossover from the diffusive regime, where the SHE occurs, to the ballistic regime, where the TMF effect appears. In fact, we found that the density $2.0\sim 3.0 \times 10^{11} \text{ cm}^{-2}$ coincides well with the value above which the ballistic negative non-local resistance (the inset of **Fig. 1e**) and TMF signals appear (**Fig. 2** and **Supplementary Fig. 4**) within the resolution of our experiment. More in-depth studies on the crossover between the diffusive spin-Hall and ballistic TMF effects or their coexistence may lead to a better understanding of the charge transport in spin-orbit-coupled systems, and our study shows that it is possible in high-quality graphene-TMDC heterostructures.”

Figure R9. The 1D cuts of TMF spectra near zero density for **a.** Sample 1 and **b.** sample 2 shown in **Fig. 2a-b**. The figure is identical to **Supplementary Fig. 4**.

Reviewers' Comments:

Reviewer #1:

Remarks to the Author:

The authors satisfactorily addressed this reviewers comments and thus I recommend the article for publication.

Reviewer #2:

Remarks to the Author:

The authors have taken the reviewer's suggestions into account and made the necessary revisions to their manuscript. They have included TMF spectra on the electron side in the supplementary information, as recommended by the reviewer. Furthermore, the authors have provided a more detailed explanation regarding the uneven heights of the focusing peaks, addressing the origin of spin polarization and acknowledging the need for further analysis and controlled experiments. They have also clarified the behavior of the first peak's splitting with increasing carrier density, considering the limitations of their device and the lack of clear experimental evidence for electric-field-tuned spin-orbit coupling (SOC) in monolayer graphene-TMDC heterostructures. The authors have made modifications to improve the clarity of their statement on the spin-valley Zeeman term and have revised the figures and text accordingly. Additionally, they have appreciated the reviewer's suggestion to present $\Delta B/B$ in Fig. 5e, enabling better comparison with previous data. Overall, the authors have adequately addressed all the reviewer's comments, including additional data where necessary, resulting in an improved manuscript. Therefore, I recommend that this manuscript be published in this esteemed journal.

Reviewer #3:

Remarks to the Author:

Reviewer report to the manuscript NCOMMS-23-16631A

I am satisfied with the authors' response and the useful data newly added in the revised manuscript and can now recommend publication of this paper. Nevertheless, there still some suggestions that can improve the manuscript.

1. In Reply 1, the authors provide the conductance data as a function of the gate voltage for both samples 1 and 2 (Fig. R5, identical to Supplementary Fig. 1a). It is worth mentioning that the WSe_2 becomes conducting, which causes obvious conductance fluctuations at the carrier density larger than $1 \times 10^{12} \text{ cm}^{-2}$ due to the charge transfer between the WSe_2 and graphene layers.

2. In Reply 2, the author proved the data for obtaining the gate oxide capacitance. The capacitance typically should be the quantity measuring the tolerance of charges per unit voltage rather than the number of charges per unit voltage. Therefore, the applied unit and the presented value have to be checked.

3. In Reply 3, the authors demonstrate that the spin-orbit proximity effect should come from the atomic potential decoration from van der Waals interactions rather than WSe_2 defects. I suggest that the comments about the proximity source in the response can be included in the manuscript.

4. In Reply 4, the authors argue that it is not straightforward to discuss the origin of spin polarization from the uneven heights of the focusing peak by comparing the experimentally measured and simulation results. However, the uneven peak heights of first split focusing peaks from the simulation can and should be explained. The authors state that "the stronger the magnetic field, the more electrons are bent, and the more electrons are collected in the detector probe, leading to enhanced transmission from the injector to the detector." This predicts a larger peak height at a large focusing field, which opposes the simulation TMF spectrum (Figs. 1d and

3f).

5. The authors use the “two-point” conductance based on the Landauer and Büttiker formalism to quantify the obtained focusing electrons into the collector in their simulation work. It causes some confusion with the case that the conceptual device is only two-terminal. However, the simulation device is actually three-terminal and the three-terminal conductance was studied. Therefore, in my opinion, whether the locally or non-locally conductance was measured is much more crucial.

To all reviewers

We sincerely appreciate the time and efforts of all three reviewers to carefully read our reply, revised manuscript, and supplementary again, and to provide some suggestions to improve our manuscript. We are very glad to learn that all referees are satisfied with our reply and changes made and recommend publication.

We have modified the manuscript and supplementary by following the suggestions made by the referee, and we believe our paper now becomes more accurate. The point-by-point reply to the comments of the referees follows here below.

Reviewer #1

Comment: *The authors satisfactorily addressed this reviewers comments and thus I recommend the article for publication.*

Reply: We are very pleased to read that the referee considers our reply satisfactory and recommends publication. We thank the referee for his/her constructive comments and remarks that have led to an improvement of our work.

Reviewer #2

Comment: *The authors have taken the reviewer's suggestions into account and made the necessary revisions to their manuscript. They have included TMF spectra on the electron side in the supplementary information, as recommended by the reviewer. Furthermore, the authors have provided a more detailed explanation regarding the uneven heights of the focusing peaks, addressing the origin of spin polarization and acknowledging the need for further analysis and controlled experiments. They have also clarified the behavior of the first peak's splitting with increasing carrier density, considering the limitations of their device and the lack of clear experimental evidence for electric-field-tuned spin-orbit coupling (SOC) in monolayer graphene-TMDC heterostructures. The authors have made modifications to improve the clarity of their statement on the spin-valley Zeeman term and have revised the figures and text accordingly. Additionally, they have appreciated the reviewer's suggestion to present $\Delta B/B$ in Fig. 5e, enabling better comparison with previous data. Overall, the authors have adequately addressed all the reviewer's comments, including additional data where necessary, resulting in an improved manuscript. Therefore, I recommend that this manuscript be published in this esteemed journal.*

Reply: We are very pleased to read that the referee acknowledges that our reply and revised manuscript have addressed all reviewers' comments adequately and the paper is now ready to be published. We thank the referee for his/her critical comments and constructive suggestions that allowed us to make the changes that improved the quality of our work.

Reviewer #3

Comment: *I am satisfied with the authors' response and the useful data newly added in the revised manuscript and can now recommend publication of this paper. Nevertheless, there still some suggestions that can improve the manuscript.*

Reply: We are very glad to learn that the referee is satisfied with our reply and recommend publication. We also thank the referee to provide both constructive comments in the first round of review and the additional suggestions in this second round to improve our manuscript further.

Comment 1: 1. In Reply 1, the authors provide the conductance data as a function of the gate voltage for both samples 1 and 2 (Fig. R5, identical to Supplementary Fig. 1a). It is worth mentioning that the WSe₂ becomes conducting, which causes obvious conductance fluctuations at the carrier density larger than $1 \times 10^{12} \text{ cm}^{-2}$ due to the charge transfer between the WSe₂ and graphene layers.

Reply 1: We thank the referee for this suggestion. We have modified the caption for Supplementary Fig. 1a accordingly (changes are highlighted in yellow),

“a The four-terminal ... saturation and large fluctuations at around $n > 1 \times 10^{12} \text{ cm}^{-2}$... The abrupt change of the conductance and large fluctuations come from the WSe₂ flake being conducting that pinches off the transport through graphene by sinking its charge carriers as found in various studies¹⁻⁵. b The Hall ...”.

Comment 2: 2. In Reply 2, the author proved the data for obtaining the gate oxide capacitance. The capacitance typically should be the quantity measuring the tolerance of charges per unit voltage rather than the number of charges per unit voltage. Therefore, the applied unit and the presented value have to be checked.

Reply 2: The referee is correct. The quantity we measure is the areal gate capacitance divided by electric charge (C_{bg}/e) with the unit [$\text{Fm}^{-2}\text{C}^{-1}$] = [$\text{m}^{-2}\text{V}^{-1}$], not the total capacitance in the unit of Farad [F]. In 2D materials research, we often state it as the (gate) capacitance because it is directly related to the charge density $n = C_{bg}/e \times \Delta V_{bg}$ with $\Delta V_{bg} \equiv V_{bg} - V_{CNP}$ (V_{CNP} : V_{bg} at the charge neutrality point) that is mostly used in the data analysis. We have revised the caption for the Supplementary Fig. 1b to clarify this point as follows (changes are highlighted in yellow),

“..., where the linear fitting gives the areal capacitance divided by electric charge, $C_{bg}/e = 0.66 \times 10^{11} \text{ cm}^{-2}\text{V}^{-1}$ with $C_{bg} = 1.056 \times 10^{-4} \text{ Fm}^{-2}$. From this value, the charge density can be obtained, $n = C_{bg}/e \times \Delta V_{bg}$ with $\Delta V_{bg} \equiv V_{bg} - V_{CNP}$ (V_{CNP} : V_{bg} at the charge neutrality point).”

Comment 3: 3. In Reply 3, the authors demonstrate that the spin-orbit proximity effect should come from the atomic potential decoration from van der Waals interactions rather than WSe₂ defects. I suggest that the comments about the proximity source in the response can be included in the manuscript.

Reply 3: We thank the referee for this suggestion. Indeed, by showing the effect of SOC in ballistic transport regime, our study further confirms that the SOC can be induced by proximity with TMDC not only by the defects. We find that the most adequate position to discuss this point is after we talk about the crossover from diffusive to ballistic regime at the page 10.

Thus, we add the paragraph below at the end of the page 10,

“Additionally, the observation of both diffusive SHE at low density and ballistic TMF peak splitting at higher density indicates that the SOC in our sample is induced by proximity with TMDC^{2,4-7} not by defects¹ as the defects would have strongly suppressed ballistic transport in the sample. It confirms that in graphene-

TMDC heterostructures—thanks to the atomically sharp interface—the atomic potentials generated by the TMDC can influence graphene band strongly to create an effective Hamiltonian with distinctive SOC terms shown in Eq. (1)^{2,4-7}. This is similar to how the atomic potentials of boron and nitrogen atoms in hBN creates the moiré minibands in graphene-hBN moiré structure⁵⁸⁻⁶¹. We note that such a proximity effect does not require charge carriers in graphene to fill the energy bands in TMDC (or hBN) and thus it can occur even when there are no defect sites in TMDC that can sink charge carriers from graphene and suppress the ballistic transport. Our observation also aligns well with other studies on similar graphene-TMDC heterostructures^{2,3,7,16,32-38,62}.”

Comment 4: *In Reply 4, the authors argue that it is not straightforward to discuss the origin of spin polarization from the uneven heights of the focusing peak by comparing the experimentally measured and simulation results. However, the uneven peak heights of first split focusing peaks from the simulation can and should be explained. The authors state that “the stronger the magnetic field, the more electrons are bent, and the more electrons are collected in the detector probe, leading to enhanced transmission from the injector to the detector.” This predicts a larger peak height at a large focusing field, which opposes the simulation TMF spectrum (Figs. 1d and 3f).*

Reply 4: We apologize for misunderstanding the previous comment 4 of the reviewer 3. The reply was about the growing trend of the simulated conductance in magnetic field not about the uneven peak heights. For the uneven peak heights, we believe that it simply comes from the shape of the TMF peak. Take the topmost curve of Fig. 1d for example. By considering the same carrier density and magnetic field range, the TMF curves with and without the spin-orbit coupling are shown here:

As seen in the blue curve without SOC, the first peak is asymmetric in magnetic field with a steeper slope at the left side (i.e., at a stronger B). When SOC is turned on, such an asymmetric shape of the peak results in the lower split peak in the red curve at the steeper side (in other words, at a larger B). We note that the same feature is also seen in Supplementary Figure 3a obtained from the ray tracing simulation that is purely classical without any quantum nature. Therefore, we conclude that the uneven peak heights of the first split focusing peaks do not seem to contain particularly interesting physics that deserves to be discussed in this work.

Comment 5: *5. The authors use the “two-point” conductance based on the Landauer and Büttiker formalism to quantify the obtained focusing electrons into the collector in their simulation work. It causes some confusion with the case that the conceptual device is only two-terminal. However, the simulation device is actually three-terminal and the three-terminal conductance was studied. Therefore, in my opinion, whether the locally or non-locally conductance was measured is much more crucial.*

Reply 5: We regret that the phrase “two-point conductance” caused confusion to the opinion of the reviewer 3. To clarify this, we have changed “two-point conductance” to “conductance” at various places in both manuscript and supplementary and added “calculated for an effective three-terminal device” in the caption of Fig. 1d to make our descriptions as clear as possible. We thank the reviewer 3 for pointing this out.

Reviewers' Comments:

Reviewer #3:

Remarks to the Author:

Reviewer report to the manuscript NCOMMS-23-16631B

All my concerns have been addressed in the current manuscript and it is now well suitable for publication in my opinion.

Reviewer #3

Comment: *All my concerns have been addressed in the current manuscript and it is now well suitable for publication in my opinion.*

Reply: We are very pleased to read that the referee is satisfied with our response and considers that the revised version is now suitable for publication. We thank the referee for his/her constructive comments and remarks that allowed us to improve our work.